# Regulation of species metabolism in synthetic community systems by environmental pH oscillations

Shubin Li[1], Yingming Zhao[1], Shuqi Wu[2], Xiangxiang Zhang[1], Boyu Yang[1], Liangfei Tian [2] ✉ & Xiaojun Han [1] ✉

Constructing a synthetic community system helps scientist understand the complex interactions among species in a community and its environment. Herein, a two-species community is constructed with species A (artificial cells encapsulating pH-responsive molecules and sucrose) and species B (*Saccharomyces cerevisiae*), which causes the environment to exhibit pH oscillation behaviour due to the generation and dissipation of $CO_2$. In addition, a three-species community is constructed with species A′ (artificial cells containing sucrose and G6P), species B, and species C (artificial cells containing $NAD^+$ and G6PDH). The solution pH oscillation regulates the periodical release of G6P from species A′; G6P then enters species C to promote the metabolic reaction that converts $NAD^+$ to NADH. The location of species A′ and B determines the metabolism behaviour in species C in the spatially coded three-species communities with CA′B, CBA′, and A′CB patterns. The proposed synthetic community system provides a foundation to construct a more complicated microecosystem.

Organisms inevitably live in communities[1,2]. The dynamic interactions between different species and their environment can improve their opportunity for survival, as species can exhibit more advanced behaviours through the cooperative operation of individual cells compared with single cells[3–5]. Understanding these complex dynamic behaviours will help us decode the operating principles of biological systems that support and maintain life; in addition, this knowledge will provide a foundation for researchers to advance future microscale technologies that exhibit key features of living systems[6–11]. Artificial cells are cell-like structures that mimic partial/whole cell structure and functions. These cells have been prepared using natural or synthetic materials[12–14] and are used to clarify the working mechanism of cells[15–22]. Through the secretion and recognition of diffusive signalling molecules in their local environment, artificial cells/live cells in synthetic communities can actively communicate with each other and their surrounding

environment to realise critical dynamic biological behaviours, such as predation[23], protein expression[24,25], motility[26,27], quorum sensing[28–30], and differentiation[31,32]. Despite the significant advances, the current synthetic communities are primitive versions of dynamic interactions in biological systems, as most of these interactions are unidirectional and complicated. Although the two-way interaction between artificial cells and bacteria has recently been reported, this interaction did not involve a dynamic feedback process[28]. In the soil microbial community, the metabolites secreted by bacteria reduce the pH value of the environment and regulate the metabolism of species in the community[33,34]. The development of endowment functional feedback systems that involve the environment in synthetic community systems remains an enormous challenge.

In this work, we establish a dynamic feedback system involving artificial cell species, biological cell species, and their environment. To

[1]State Key Laboratory of Urban Water Resource and Environment, School of Chemistry and Chemical Engineering, Harbin Institute of Technology, Harbin 150001, China. [2]Key Laboratory of Biomedical Engineering of Ministry of Education, Zhejiang Provincial Key Laboratory of Cardio-Cerebral Vascular Detection Technology and Medicinal Effectiveness Appraisal, Department of Biomedical Engineering, Zhejiang University, Hangzhou 310027, China. ✉e-mail: liangfei.tian@zju.edu.cn; hanxiaojun@hit.edu.cn

this end, an environmentally responsive artificial cell (species A) that show different secretion capacities at different pH values is first established. Through the secretion of diffusive molecules, environmentally responsive artificial cells can communicate with biological cells (*Saccharomyces cerevisiae*, species B), leading to a decrease in pH in their environment. Consequently, the decrease in environmental pH can reduce the secretion capacity of species A, leading to an increase in environmental pH. Using this dynamic feedback system, we establish a stable oscillation synthetic community system, which control the enzymatic reaction in another artificial cell (species C). Finally, we explore the influence of spatial location on the communication process of the three species. Introducing a feedback system to simulate the interaction between community and environment lay a foundation for the design and clarification of complex systems in nature.

## Results

### Construction of sucrose-containing pH-responsive artificial cells (species A)

The pH-response property of sucrose-containing artificial cells (species A) originated from the pH-responsive molecules (Supplementary Fig. 1) inside giant unilamellar vesicles (GUVs). Although there are many pH-responsive hydrogelation molecules[35–39], the pH-responsive molecule here possesses fast response properties and a suitable pH-response range. The pH-responsive molecules changed from the fluid phase at high pH values (≥7) into the gel phase at low pH values (≤6) due to hydrogen bonds (Fig. 1a and b). ATR-FTIR data confirmed the presence of hydrogen bonds in the hydrogels (Supplementary Fig. 2). At pH 8, the peak at 1550 cm⁻¹ is caused by C-N stretching and N-H bending[40], and the peak at 1400 cm⁻¹ corresponds to the stretching vibration of the carboxyl group[41]. At pH 5, these two peaks shifted to 1540 cm⁻¹ and 1390 cm⁻¹, respectively, which results from the formation of hydrogen bonds among molecules in gel phase[42]. Melittin is a 26-amino-acid α-helical peptide that forms pores on the lipid bilayer membrane for substance exchange between internal and external lipid vesicles[43,44]. Here, due to the melittin nanopores on the bilayer membrane, protons could exchange between inner artificial cells and external environments. The decrease in solution pH outside artificial cells caused pH-responsive molecules inside GUVs to partially cross-link and increase the internal viscosity of artificial cells, and vice versa (Fig. 1c).

The viscosity inside artificial cells was influenced by the solution pH, as demonstrated by single particle tracking experiments. The

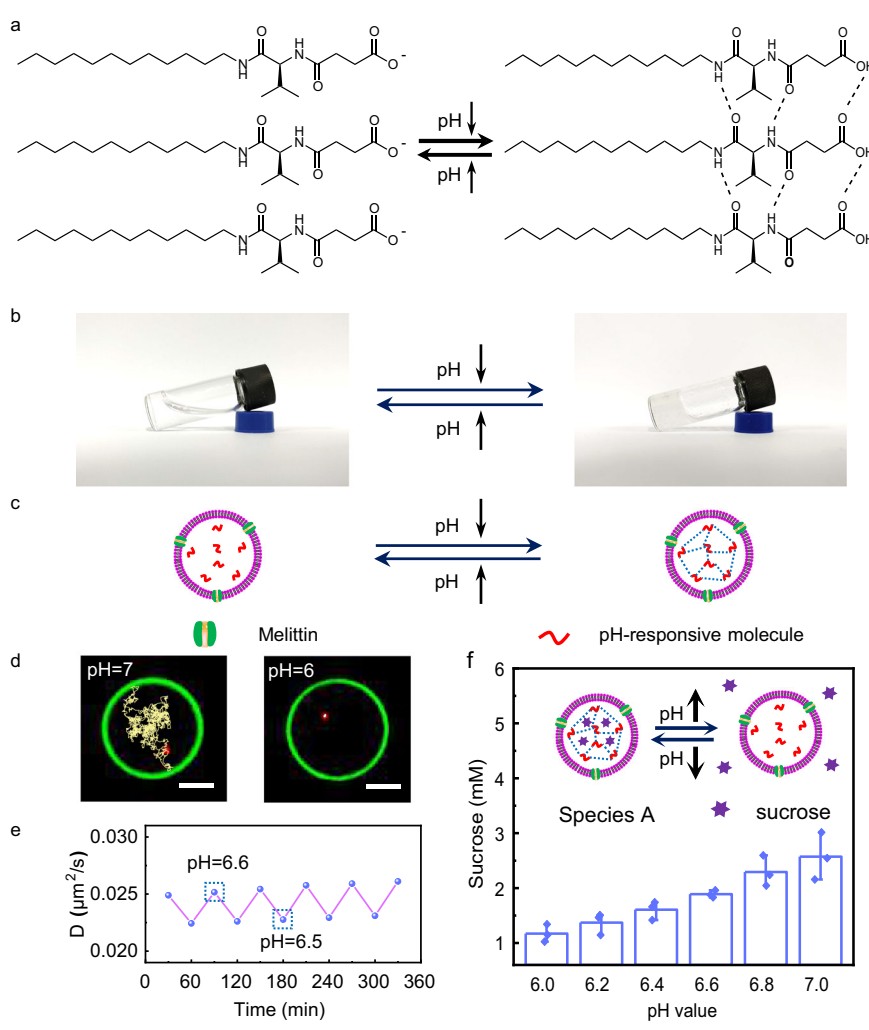

**Fig. 1 | Sucrose release property of pH-responsive artificial cells. a** Schematic diagram showing the fluid and gel phase of pH-responsive molecules against solution pH. **b** Photos of the fluid phase (pH 10) and gel phase (pH 5). **c** Schematic illustration showing the responsive property of species A. **d** Motion trajectories of red fluorescent polystyrene microspheres at pH 7 (left image) and pH 6 (right image). Scale bar is 5 μm. **e** Oscillation of the inner solution diffusion coefficient of pH-responsive artificial cells as a function of time by alternating the solution pH between 6.5 and 6.6. **f** Sucrose leakage from pH-responsive artificial cells 20 min after melittin was added as a function of solution pH. The release concentration of sucrose is obtained from three independent samples. Data are presented as the mean values ± SDs, *n* = 3. Source data are provided as a Source Data file.

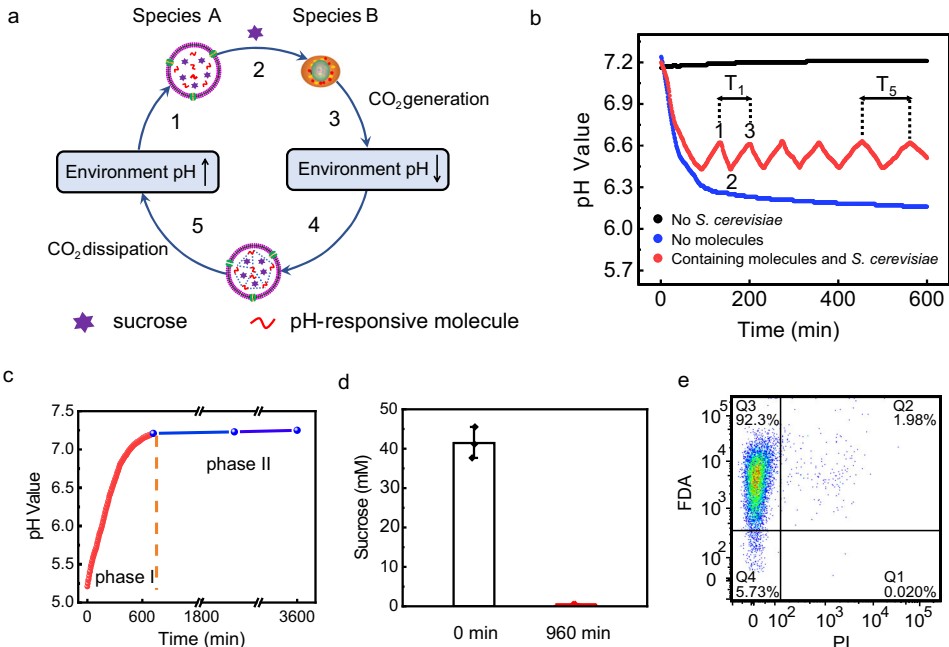

**Fig. 2 | Construction of a two-species community of pH-responsive artificial cells (containing sucrose) (species A) and *S. cerevisiae* (species B) and its pH oscillatory environment. a** Schematic illustration of solution pH oscillation caused by feedback between species A and B. **b** A typical solution pH oscillation of a two-species-community system as a function of time. Species A and B were $2.76 \times 10^6$/mL and $4.36 \times 10^6$/mL, respectively. **c** pH value of $CO_2$ oversaturated (phase I, red curve) and saturated (phase II, blue curve) gadobutrol solution as a function of time. $CO_2$ was injected into gadobutrol solution in phase I. The gadobutrol solution was saturated by $CO_2$ from air in phase II. **d** The initial sucrose concentration and sucrose concentration at the end of oscillation after 960 min. The sucrose concentrations were obtained by adding 10% Triton X-100 into the solution to release sucrose from species A. The concentration of sucrose at initial and end (960 min) of oscillation was tested with three independent samples. Data are presented as the mean values ± SDs, $n = 3$. **e** Flow cytometry scatter plots of live-dead stained species B at the longest oscillation time (960 min). Live *S. cerevisiae* (species B) was stained with FDA (green, in Q3), and dead *S. cerevisiae* was stained with PI (red, in Q1). The total number of particles counted was 10,000. Source data are provided as a Source Data file.

typical single particle trajectory in a fluid phase inside GUV (Fig. 1d, left image, pH 7) was significantly different from that in a gel phase (Fig. 1d, right image, pH 6). The viscosities of artificial cells encapsulating 10 mmol/L pH-responsive molecules were determined to be $5.19 \times 10^{-4}$ mPa·s at pH 10, $6.68 \times 10^{-4}$ mPa·s at pH 9, $1.47 \times 10^{-3}$ mPa·s at pH 8, $5.09 \times 10^{-3}$ mPa·s at pH 7, $149.72 \times 10^{-3}$ mPa·s at pH 6 and $173.17 \times 10^{-3}$ mPa·s at pH 5 (Supplementary Fig. 3). To investigate the pH sensitivity of artificial cells, the diffusion coefficient (D) of the internal solution was monitored using the fluorescence recovery after photobleaching (FRAP) technique by varying pH 6.5 and 6.6 at a time interval of 30 min (Fig. 1e, Supplementary Fig. 4). Distinct D value oscillation was obtained, which implied that even a 0.1 pH difference had a significant impact on the viscosity inside artificial cells. In addition, the D value did not change significantly after 5 cycles at pH 6.5 or 6.6, indicating that pH-responsive molecules remained inside artificial cells without leakage.

Taking advantage of this property, the sucrose (300 mM) inside artificial cells (species A) was released in a controlled manner by varying the solution pH (Fig. 1f). The released sucrose concentration outside the artificial cell was $1.17 \pm 0.16$ mM at pH 6.0, $1.37 \pm 0.20$ mM at pH 6.2, $1.61 \pm 0.17$ mM at pH 6.4, $1.89 \pm 0.065$ mM at pH 6.6, $2.29 \pm 0.28$ mM at pH 6.8, and $2.57 \pm 0.43$ mM at pH 7.0 20 min after melittin was added. The sucrose release rate increased with increasing solution pH.

## Construction of a two-species community (species A and B) and its pH oscillatory environment

An auto-oscillation two-species-community system was built by combining sucrose-containing pH-responsive artificial cells (species A) and *S. cerevisiae* (species B). At high environmental solution pH, the inner species A is fluid to release sucrose molecules, which are consumed by

*S. cerevisiae* (species B) to generate $CO_2$ to decrease the environmental solution pH (Fig. 2a, steps 1–3). Under these conditions, species A stops releasing sucrose (Fig. 2a, step 4), while the environmental solution dissipates $CO_2$ into the air to increase the environmental pH (Fig. 2a, step 5). The cycles repeat and cause the solution pH to undergo auto-oscillation, which was confirmed by the experimental results. With populations of species A and B of 38.8% and 61.2%, respectively (Supplementary Figs. 5 and 6), the initial solution pH of the two-species-community system was 7.2 (Fig. 2b). At this pH value, sucrose molecules were released from species A to be consumed by species B ($4.36 \times 10^6$/mL) to produce $CO_2$. Consequently, the solution pH decreased to 6.43 due to the production of protons, which further decreased the pH inside artificial cells to inhibit the release of sucrose from species A. At this stage, $CO_2$ was oversaturated in the solution because the pH of the $CO_2$ (from air)-saturated community system was confirmed to be 7.25 after 60 h of monitoring (Fig. 2c, phase II, blue curve). Therefore, $CO_2$ tended to dissipate to the air, consequently increasing the solution pH. The release of sucrose from species A due to the pH increase was in turn promoted to trigger the production of $CO_2$ by *S. cerevisiae* (species B). Thus, the auto-oscillation of solution pH was established (Fig. 2b, red curve).

For the typical oscillation period $T_1$ (Fig. 2b), $CO_2$ production dominated in the system and decreased the solution pH during the first half period (from point 1 of pH 6.63 to point 2 of pH 6.42). The dissipation of $CO_2$ to the air dominated in the system and increased the solution pH due to the inhibition of sucrose release from species A during the second half period (from point 2 of pH 6.42 to point 3 of pH 6.62). Multiple cycles of solution pH oscillations were observed. The solution pH did not oscillate in the absence of species B (Fig. 2b, black curve) or pH-responsive molecules in species A (Fig. 2b, blue curve), which confirmed that the feedback between these two species in the

community caused environmental pH oscillations. Notably, the period gradually increased as the oscillation continued (Fig. 2b, red curve). The periods of $T_1$ and $T_5$ were 68 min and 109 min, respectively. This can be explained by the consumption of sucrose inside species A, which caused the concentration of solution components to differ at each equilibrium point in subsequent cycles. The decrease in the sucrose concentration difference between the inner and outer membranes slowed the production of $H^+$ and the release of $CO_2$, thereby generating a longer oscillation period in the subsequent cycles. The longest oscillation occurred for 960 min because sucrose in species A was completely consumed (Fig. 2d, Supplementary Fig. 7). The *S. cerevisiae* was confirmed to be alive during oscillations (Fig. 2e, Supplementary Figs. 8, 9).

### Adjusting the oscillation parameters in the two-species community by changing the initial conditions

Four oscillation parameters were defined (Supplementary Fig. 10a), including the initial pH drop time ($t_0$), which was defined as the time from the beginning to the first lowest pH point, the steady median pH value, which was defined as the median of the highest and lowest pH value during the oscillation, amplitude (A) of the oscillation and period (T) of oscillation.

The solution pH oscillation behaviour was influenced by the concentration of sucrose in species A and the ratio between species A and B. Both of these factors affect the production speed of $CO_2$ by *S. cerevisiae*. When the population ratio of species A ($2.76 \times 10^6$/mL) to B ($4.36 \times 10^6$/mL) was maintained at 0.63:1, the initial concentration of sucrose in species A had a dramatic impact on the oscillation behaviours (Supplementary Fig. 10b and 10c). With the increase in the initial concentration of sucrose, the $t_0$ values almost linearly decreased from 206 min at 50 mM sucrose to 107 min at 250 mM sucrose (Supplementary Fig. 10d, blue plot), while the steady median pH values shifted from 6.96 at 50 mM sucrose to 6.62 at 250 mM sucrose with a linear decline (Supplementary Fig. 10d, pink plot). In addition, the amplitude of oscillation increased (Supplementary Fig. 10e, green plot), but the period of oscillation decreased (Supplementary Fig. 10e, purple plot). The higher sucrose in species A enabled a faster release rate to the solution, consequently resulting in a faster $CO_2$ production rate, which explained the shorter $t_0$ and the first lowest pH point. The balance between the $CO_2$ dissipation rate and $CO_2$ production rate caused a lower steady median pH value, larger amplitude and shorter period of oscillation at higher sucrose concentrations in species A. The $CO_2$ dissipation rate was faster at lower pH, since the slopes from the first lowest point to the first highest point were larger at higher sucrose concentrations (Supplementary Fig. 10b, purple dashed boxes) and the tangents of the points in the red curve ranging from 6.5 to 7.2 at low pH values (Fig. 2c, phase I) were larger. This results from the larger bias from the steady pH of the system (7.25). The faster $CO_2$ dissipation rate balanced the $CO_2$ production rate, which explained the lower steady median pH value, the larger amplitude and shorter average period at higher sucrose concentrations in species A. The gradual consumption of sucrose from species A caused pH oscillation to disappear since the $CO_2$ production rate was not enough to overcome the $CO_2$ dissipation rate. To confirm the necessity of pH-responsive release of sucrose from species A to the oscillation phenomenon, the solution pH variation of direct mixing sucrose with *S. cerevisiae* was measured (Supplementary Fig. 11). The sucrose concentrations leaking out from species A (containing 50 mM, 100 mM, 150 mM, 200 mM, 250 mM, 300 mM sucrose) for 30 min (Supplementary Fig. 11a) were chosen to mix with *S. cerevisiae*. No solution pH oscillations were observed (Supplementary Fig. 11b). The solution pH dropped in the first phase and gradually increased in the second phase.

When the number of species B in the community was maintained, the number of species A (containing 150 mM sucrose) also had a dramatic impact on the oscillation behaviours (Supplementary Figs. 12,

13a). With the increase in the percentage of species A, the $t_0$ values decreased from 93 min at the ratio of species A to B (1.4:1) to 45 min at the ratio of species A to B (15.0:1) (Supplementary Fig. 13b), while the steady median pH values shifted from 6.62 at the ratio of species A to B (1.4:1) to 6.14 at the ratio of species A to B (15.0:1) (Supplementary Fig. 13c). In addition, the amplitude of oscillation increased, but the period of oscillation decreased (Supplementary Fig. 13d). The higher percentage of species A in the community apparently increased the initial sucrose concentration in species A, which resulted in a faster $CO_2$ production rate by *S. cerevisiae*, further causing a shorter $t_0$ and the first lowest pH point. Similar to the results obtained by increasing the initial sucrose concentration, the balance of the $CO_2$ dissipation rate and $CO_2$ production rate caused a lower steady median pH value, larger amplitude and shorter period of oscillation at a higher percentage of species A.

The construction of a two-species-community oscillation system helps researchers construct complex dynamic response systems using simple components. The dynamic interaction among species A, species B and the environment maintained the pH stability of the system, which may be beneficial to the survival of pH-sensitive species. Although two-species communities are a useful platform for studying interspecies interactions, natural communities typically consist of more than two species to maintain community stability. In the following content, a three-species community was constructed to investigate the interactions among species and the environment.

### Construction of three-species communities (species A', B and C) and their interactions

In the abovementioned two-species-community system, we demonstrated that the solution (environment) pH oscillation was controlled by the feedback between species A and B. To further explore whether this behaviour influences the metabolism of the third species, a more complicated community was built with pH-responsive artificial cells containing sucrose and G6P (species A'), *S. cerevisiae* (species B) and artificial cells containing $NAD^+$ and G6PDH (species C) (Fig. 3a, Supplementary Fig. 14). The controlled release of G6P from species A' was expected to trigger the stepped generation of NADH inside species C (Fig. 3a). Glucose-6-phosphate dehydrogenase (G6PDH) is an important enzyme involved in the glycolysis pathway. Upon receiving G6P, the G6PDH inside species C converted G6P and $NAD^+$ to 6-phosphate gluconate lactone and NADH, respectively. NADH is an important biomolecule that provides reductive force in many metabolic pathways. Species C is an artificial cell that can perform metabolism mimicry.

The solution pH oscillation of the three-species-community system showed similar behaviour to the two-species-community system with the same amount of species A and B. The addition of species C did not affect the solution pH oscillation behaviour (Fig. 3b). For the typical oscillation period $T_1$ (Fig. 3b), the release rate of G6P and sucrose from species A' gradually decreased in the first half of the period (from point 1 to point 2). The released G6P diffused into species C through melittin pores in the bilayer membrane to react with $NAD^+$ to generate NADH molecules. The production of NADH molecules was monitored by a fluorescence spectrometer since NADH emits blue light (460 nm) (Supplementary Fig. 15). Corresponding to the first half period (from point 1 to point 2, Fig. 3b), the intensity of the three-species-community system slowly increased, as shown in Fig. 3c (from point 1 to 2), which implied that NADH was generated inside species C. To further demonstrate the generation of NADH inside species C, the three-species-community system was monitored by a fluorescence microscope. The fluorescence images showed a three-species community (Fig. 3d), including species A (labelled with green NBD-PE in bilayer, first column image of each row), species B (bright field image, second column image of each row), and species C (blue GUVs with no

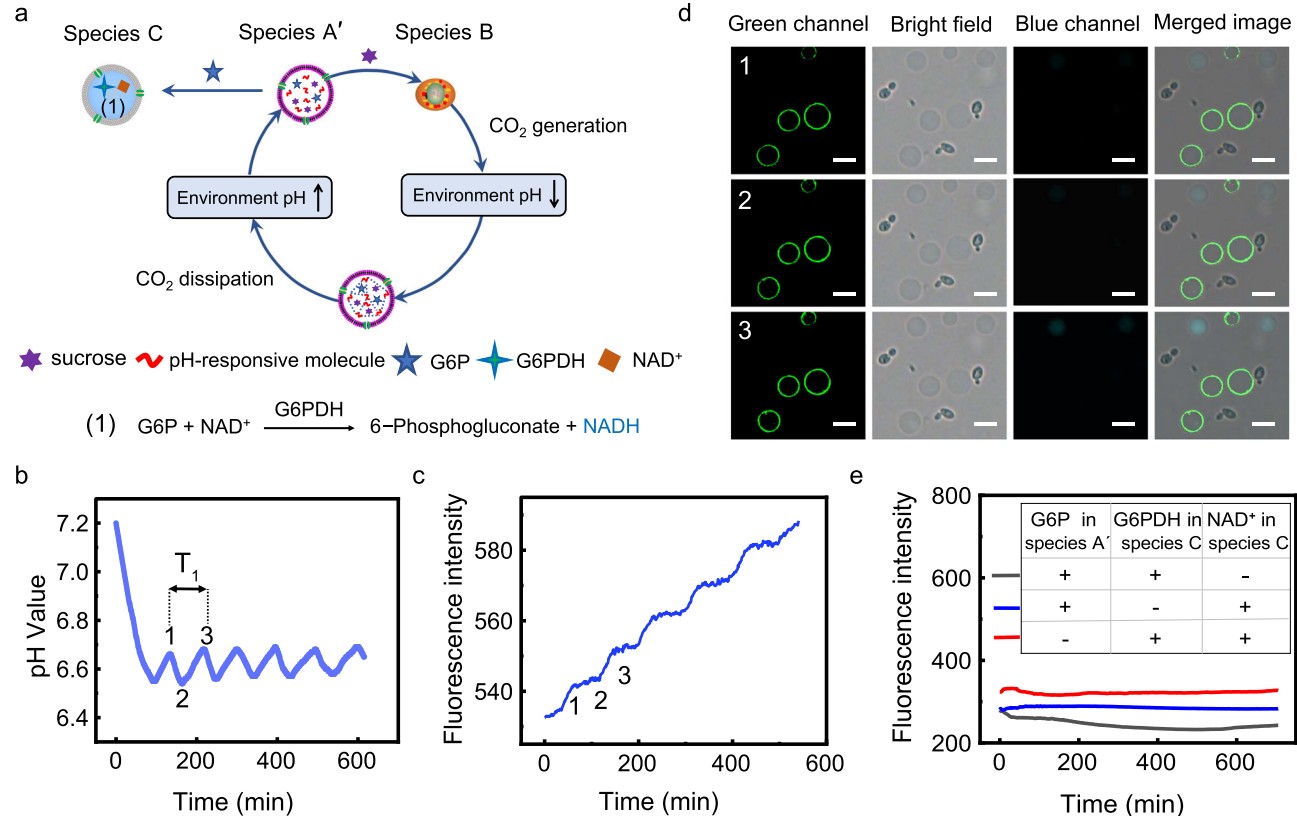

**Fig. 3 | Three-species community containing sucrose and G6P containing pH-responsive artificial cells (species A'), _S. cerevisiae_ (species B) and NAD⁺ and G6PDH containing artificial cells (species C). a** Schematic diagram showing solution pH oscillation and NADH generation inside species C caused by the feedback between species A' and species B. **b** A typical solution pH oscillation of a three-species-community system as a function of time with species A' ($6.10 \times 10^6$/mL) containing 150 mM sucrose and 10 mM G6P, species B ($4.36 \times 10^6$/mL), and species C ($4.36 \times 10^6$/mL) containing 10 mM NAD⁺ and 10 μg/mL G6PDH.
**c** Fluorescence intensity of the same community as (**b**) as a function of time.
**d** Corresponding microscopy images of the community system at the time points in (**b**). The first column images were observed using a green filter for viewing species A

'. The second column images were observed in bright field for viewing species B. The third column images were recorded using a blue filter for viewing species C. The fourth column images were the merged images of Columns 1, 2 and 3 of each row. Scale bar is 5 μm. **e** The fluorescence intensity of the same three-species communities as a function of time with the absence of G6PDH in species C (blue curve), G6P in species A' (red curve), or NAD⁺ in species C (black curve). G6PDH and G6P are the abbreviations of glucose-6-phosphate dehydrogenase and glucose-6-phosphate, respectively. NAD⁺ and NADH are the abbreviations of nicotinamide adenine dinucleotide and nicotinamide adenine dinucleotide plus hydrogen. Source data are provided as a Source Data file.

label in bilayer, third column image of each row). The average intensity in species C (Supplementary Fig. 16) exhibited a similar stepwise increase to the fluorescence spectroscopy data (Fig. 3c). Therefore, NADH generation was confirmed inside species C. During the second half period (from point 2 to 3), the release rate of G6P gradually increased, which caused more G6P to diffuse into species C; as a result, a fast increase in intensity was observed (Fig. 3c, point 2 to 3; Supplementary Fig. 16, point 2 to 3). The slow and fast generation of NADH was repeated along with solution pH oscillation, which caused a stepwise increase in fluorescence intensity (Fig. 3c, Supplementary Fig. 16). No NADH was generated inside species C with the absence of G6PDH in species C (Fig. 3e, blue curve), NAD⁺ in species C (Fig. 3e, black curve) or G6P in species A' (Fig. 3e, red curve), which confirmed that the feedback between species A' and species B regulated the internal metabolism in species C. In the following context, this phenomenon was used to investigate the effect of species spatial distribution on their communications.

The spatial distribution of species in communities is important for communication among species and for species to execute their functions. Spatially coded three-species communities were constructed inside each well of a steel mesh (Supplementary Fig. 17) under a magnetic field using the Magneto-Archimedes principle with a homemade device[45,46] (Supplementary Fig. 18). Gadobutrol is a

paramagnetic contrast agent used in magnetic resonance imaging[47]. Gadobutrol was added to increase magnetic susceptibility of solution. Three patterns (CA'B, CBA', A'CB) of the three-species community were obtained by varying the addition order of species A', species B, and species C (Supplementary Fig. 19). The projected confocal images of the three-species communities in a typical well (Supplementary Figs. 20 and 21) confirmed the formation of communities with the species order CA'B, CBA', and A'CB. By monitoring the solution pH, it was determined that the community with CA'B and CBA' patterns maintained oscillation behaviour (Fig. 4a and d), while the community with A'CB lost its capability to oscillate solution pH (Fig. 4g). In the A'CB community, closely packed species C inhibited the feedback between species A' and species B, which disabled the pH oscillation capacity of the community. A laser scanning confocal microscope was used to monitor NADH generation inside species C in spatially coded communities as a function of time. The images in one typical period (points 1, 2 and 3) of each spatially coded community corresponding to Fig. 4a, d and g were obtained in Fig. 4b, e, and h, respectively. The fluorescence intensity curves confirmed the stepped generation of NADH in species C in the community of CA'B (Fig. 4c), which was consistent with their pH oscillation behaviour. The intensity of community CBA' (Fig. 4f) was almost unchanged because closely packed species B inhibited G6P diffusion into species C. The intensity of

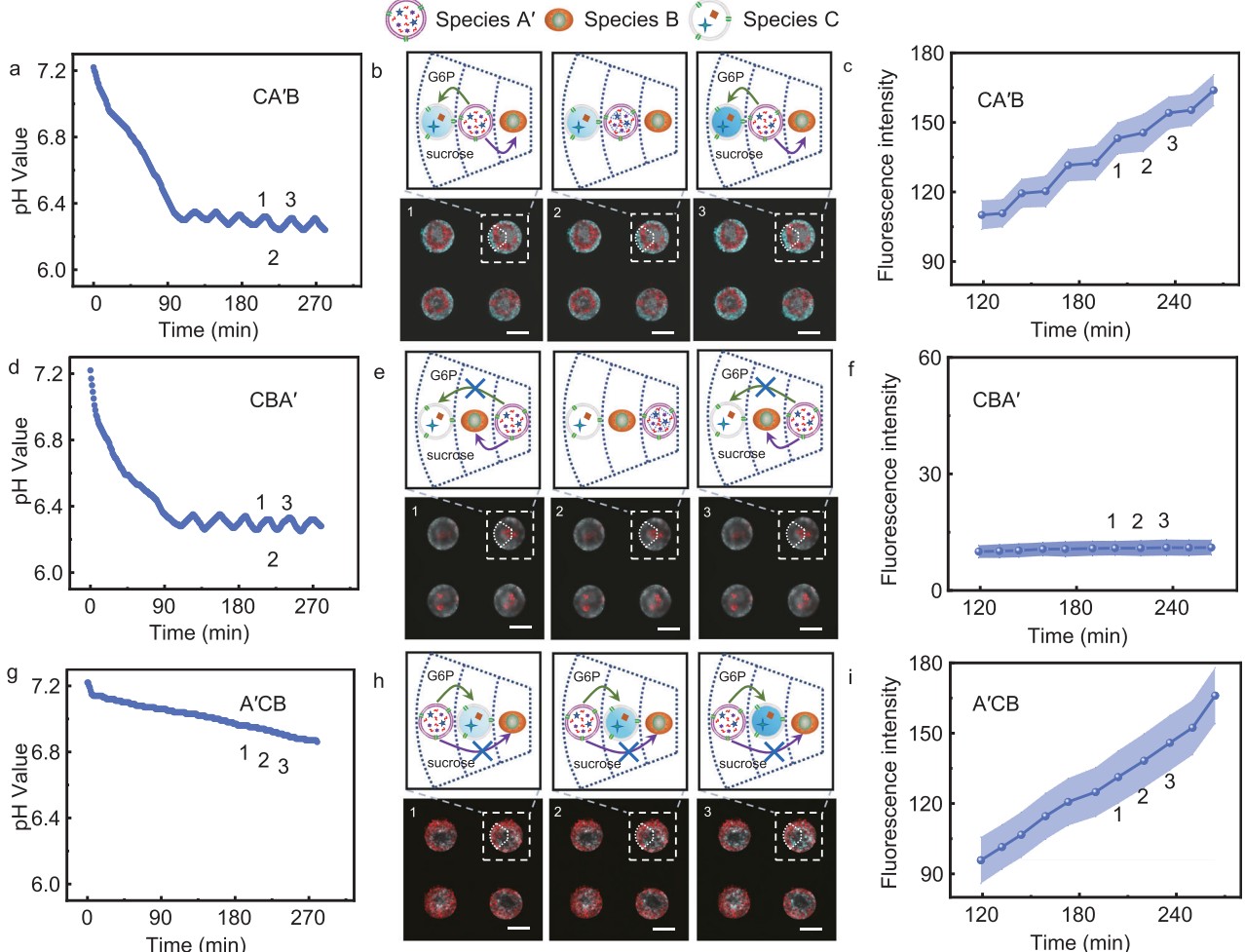

**Fig. 4 | Communication among spatially coded three-species communities.**
**a** The pH oscillation curve of community CA'B as a function of time. **b** The corresponding laser scanning confocal microscopy images at points 1, 2, and 3 in (**a**). **c** The fluorescence intensity of species C of community CA'B as a function of time. **d** The pH oscillation curve of CBA' as a function of time. **e** The corresponding laser scanning confocal microscopy images at points 1, 2, and 3 in (**d**). **f** The fluorescence intensity of species C of community CBA' as a function of time. **g** The pH oscillation curve of A'CB as a function of time. **h** The corresponding laser scanning confocal microscopy images at points 1, 2, and 3 in (**g**). **i** The fluorescence intensity of species C of community A'CB as a function of time. The mean fluorescence intensities of species C in (**c**), (**f**), and (**i**) are from ten independent samples. Data are presented as the mean values ± SDs, $n = 10$. G6P is the abbreviations of glucose-6-phosphate. Source data are provided as a Source Data file.

community A'CB (Fig. 4i) continuously increased because the G6P released from species A' at relatively high pH values (Fig. 4g) diffused into adjacent species C to generate NADH. To observe the inhibitory effect of tightly packed species on the diffusion of small molecules more intuitively, we prepared species A″ encapsulating a model dye molecule of fluorescein (Mw = 332.3 g/mol), which exhibits green fluorescence and a molecular weight similar to sucrose (Mw = 342.3 g/mol). Fluorescein leakage was observed from individual species A″ (Supplementary Fig. 22) and spatially patterned A″ (Supplementary Fig. 23). Two three-species communities were constructed by arranging species A″ (Supplementary Fig. 24a), species B (Supplementary Fig. 24b), and species C (labelled with TR-DHPE, Supplementary Fig. 24c) in the order of A″CB (Supplementary Fig. 24d–f) and CBA″ (Supplementary Fig. 24h–j). No green fluorescence was observed in the species B region in the A″CB community within 270 min (Supplementary Fig. 24g) due to the close pack of species C. Similarly, no green fluorescence was observed in the species C region due to the close pack of species B (Supplementary Fig. 24k). All the above-mentioned results demonstrated that the spatial distribution of species in the community exhibited a dramatic impact on the behaviour of the community system.

## Discussion

The construction of synthetic communities plays an important role in research on interaction between species and the environment or research to clarify the complex behaviour among populations. Three artificial cells are built by encapsulating sucrose/pH-responsive molecules (species A), sucrose/pH-responsive molecules/G6P (species A'), and NAD⁺/G6PDH (species C). *S. cerevisiae* is species B. A two-species community is constructed using species A and species B, which causes the pH of its environment to oscillate by the balance of sucrose consumption by *S. cerevisiae* and $CO_2$ dissipation from solution. The sucrose released from species A is consumed by *S. cerevisiae* (species B) to generate $CO_2$ to decrease the solution pH, while the oversaturated $CO_2$ dissipates into the air to increase the solution pH. The pH oscillation behaviour is influenced by the initial sucrose concentration in species A and the ratio between species A and B. A higher initial sucrose concentration causes lower steady median pH values, a larger amplitude and a shorter period. The higher ratio of species A to B results in lower steady median pH values, larger amplitudes and shorter periods. Significantly, the pH oscillation between species A' and B in a three-species-community system regulates the periodical release of G6P from species A', which tunes the metabolic reaction from NAD⁺ to

NADH inside species C. The location of species in the spatially coded three-species-community system is confirmed to have a dramatic impact on the interaction among species and between the community and its environment. The species in natural communities often occur in spatial order. The influence study of spatial distribution on signal transmission among species in spatially coded synthetic communities helps researchers clarify the structure and function of communities.

Signal communication among species in natural communities exhibits dynamic characteristics. By responding to environmental changes or signal molecules secreted by other species, species in the community can regulate their metabolism to better adapt to the environment and improve their survival ability. Two types of artificial cells were developed with the functions of environmental stimulus response (species A) and metabolism mimicry (species C). The environment, as a key element, was dynamically involved in the system to influence the function of species in constructed communities. Complex interactions among species and the environment were demonstrated in the two-species/three-species communities. Therefore, we established communities that possess complex dynamic interaction networks, which provide a way to investigate complicated internetwork reactions among microecosystems; in addition, the results lay the foundation for building more complex systems in the future to perform higher-order functions.

## Methods

### Materials
1-Palmitoyl-2-oleoyl-glycero-3-phosphocholine (POPC), cholesterol (Chol), Texas Red-labelled 1,2-dihexadecanoyl-sn-glycero-3-phosphoethanolamine triethylammonium salt (TR-DHPE) and 1,2-dioleoyl-sn-glycero-3-phosphoethanolamine-N-(7-nitro-2-1,3-benzoxadiazol-4-yl) (NBD-PE) were obtained from Avanti Polar Lipids (USA). Melittin, sucrose, galactose, and lactose were purchased from Sigma Aldrich (China). Anhydrous sodium carbonate ($Na_2CO_3$), sodium bicarbonate (NaHCO$_3$), polyethylene glycol (PEG-20000), glucose-6-phosphate (G6P), disodium hydrogen phosphate, citric acid, fluorescein, fluorescein diacetate (FDA), propidium iodide (PI), and red fluorescent monodisperse polystyrene microspheres were purchased from Aladdin (China). YM medium was obtained from Sangon Biotech (China). Fluorescein isothiocyanate-labelled bovine serum albumin (FITC-BSA), oxidised coenzyme I (NAD$^+$), and glucose-6-phosphate dehydrogenase (G6PDH) were obtained from Solarbio (China). The NAD$^+$ assay kit (AKCO001C) was purchased from Boxbio (China). A sucrose assay kit (A099-1-1) was purchased from Nanjing Jiancheng Bioengineering Institute (China). The stainless steel (SS) mesh (1.0 cm × 1.0 cm) with a thickness of 100 μm and a hole diameter of 300 μm was custom-made by RGRS Company (China). Cylindrical NdFeb magnets (1 T, diameter = 3 cm, thickness = 1 cm) were obtained from Gates Qiangci Company (China). The experimental water was obtained from Milli-Q IQ 7005 (Germany).

### General procedure for synthesising pH-responsive molecules
THF (150 mL) solution containing carbobenzyloxy-L-amino acid (5Val) (40 mmol) and N-hydroxysuccinimide (40 mmol, 1.0 equivalent) was added to a THF (75 mL) solution of N,N'-dicyclohexylcarbodiimide (10.9 mmol, 1.01 equivalent) with a drip funnel under N$_2$ at 0 °C. The mixture was further stirred at 0 °C for 1 h, followed by filtering under reduced pressure. The crude residue was purified by crystallisation in isopropanol to obtain carbobenzyloxy-L-amino activated ester. At room temperature, the solution of activated carbobenzyloxy-L-amino ester (36.8 mmol) in THF (200 mL) was added to the THF (100 mL) solution of n-dodecylamine (40.5 mmol, 1.1 equivalent) with a drip funnel under N$_2$. The mixture was further stirred at 55 °C for 5 h. After that, the mixture was cooled to room temperature, and the solvent was removed under reduced pressure. The residue was washed with water until pH = 7 and dried overnight under reduced pressure at 50 °C. The

palladium catalyst (10% w/w) was suspended in MeOH (250 mL) and stirred at room temperature for 10 min under H$_2$. Subsequently, a methanol solution of carbobenzyloxyamino compound (150 mL) was added through a syringe and stirred at room temperature for 2–4 h under H$_2$. A solution of carbobenzyloxyamino compound (16.3 mmol) in THF (150 mL) was treated with solid K$_2$CO$_3$ (61.9 mmol, 3.8 equivalent) under N$_2$ at 0 °C. The mixture was stirred at 0 °C for 15 min, and succinic anhydride was added (32.6 mmol, 2.0 equivalent). The mixture was further vigorously stirred at room temperature for 16 h. After that, the solution was concentrated under reduced pressure and dissolved in water (100 mL). Hydrochloric acid was dropped at 0 °C until the formation of white precipitates. The precipitates were filtered under vacuum, and the residue was washed with water (300 mL). The compound was dried overnight at 50 °C under reduced pressure. The compound (Supplementary Fig. 1) was a pH-responsive molecule ((S)-4-((1-(dodecylamino)-3-methyl-1-oxobutan-2-yl)amino)-4-oxobutanoic acid) with a yield of 95%.

### Viscosity measurements inside artificial cells at different pH values
Particle tracking was used to determine changes in viscosity inside the giant unilamellar vesicles (GUVs). GUVs were prepared by an emulsion method. In brief, 20 mg POPC and cholesterol (70:30, w/w) were dissolved in 80 μL chloroform. Liquid paraffin (2 mL) was added to the above chloroform solution. The mixture was heated at 80 °C for 120 min to remove chloroform from the solution. The obtained solution was cooled to room temperature as an oil phase solution. The aqueous solutions contained 300 mM sucrose, 50 mM Na$_2$CO$_3$, 0.05 g/mL PEG, 0.025% (w/v) red fluorescent polystyrene microspheres, and different concentrations of pH-responsive molecules (0.5 mmol/L, 1 mmol/L, 5 mmol/L, 10 mmol/L). The oil phase solution and the aqueous solution (10:1, v/v) were mixed and vortexed for 60 s to obtain a water-in-oil emulsion. The above emulsion was carefully added to a 1.5 mL centrifuge tube containing 250 μL isotonic galactose solution, followed by centrifugation for 30 min (×10,000 g). The resulting precipitate was GUVs, which were regarded as artificial cells in this paper.

The buffer solutions with different pH values (pH = 5, 6, 7, 8) were prepared by mixing 0.2 mol/L disodium hydrogen phosphate with 0.1 mol/L citric acids at different volume ratios. The buffer solutions (pH = 9, 10) were prepared by mixing 0.1 mol/L Na$_2$CO$_3$ with 0.1 mol/L NaHCO$_3$ at different volume ratios. The artificial cell solutions (1 μL) were added to buffers with different pH values (29 μL); subsequently, melittin (1 μg/mL) was added to the solution to help balance the pH inside and outside the artificial cells. Time-dependent movements of individual fluorescent microspheres trapped within artificial cells were monitored by capturing a series of images at intervals of 160 ms using a fluorescence microscope (Olympus IX73, Japan). The diffusion constants of the microspheres (1 μm in diameter) were calculated from the 2-D projections of particle movements in the x and y directions using Eq. 1:

$$r_{RMS} = \frac{\sum_1^{n-1}\sqrt{(x_{n+1} - x_n)^2 + (y_{n+1} - y_n)^2}}{n - 1} \quad (1)$$

where $r_{RMS}$ is the root mean square distance travelled by the microspheres during a set time t. The diffusion constant (D) was calculated using Eq. 2:

$$D = \frac{r_{RMS}^2}{4\Delta t} \quad (2)$$

where $\Delta t$ is 160 ms. The viscosity inside the artificial cells was obtained according to the Einstein–Stokes Eq. 3:

$$\eta = \frac{k_b T}{6\pi r_p D} \quad (3)$$

where $k_b$ is the Boltzmann's constant, T is the temperature, and $r_p$ is the radius of the microspheres.

## Fluorescence recovery after photobleaching (FRAP) experiment

The artificial cell solutions (1 μL) were added to isotonic buffers (29 μL, pH = 6.6), and melittin (1 μg/mL) was subsequently added to the solution to observe the recovery of FITC-BSA fluorescence bleaching in artificial cells. The external solution of artificial cells was replaced with isotonic buffer solution (pH = 6.5) after 30 min. The above procedure was repeated 10 times.

FRAP experiments were performed using a laser scanning confocal inverted microscope (Olympus FV3000, Japan). The fluorescence recovery of bleached spots was monitored inside artificial cells with a bleaching radius of r = 0.9 μm. The apparent diffusion coefficient (D) in the GUV was calculated using Eq. 4:

$$D \approx r^2/\tau \tag{4}$$

where $\tau$ is the recovery time.

## Sucrose release experiment

The prepared artificial cells containing 10 mmol/L pH-responsive molecules, sucrose (50 mM, 100 mM, 150 mM, 200 mM, 250 mM, 300 mM) and 0.05 g/mL PEG-20000 were suspended in 500 μL isotonic solutions of different pH values with the addition of melittin (1 μg/mL). Then, the supernatant was collected by centrifugation at different time points. Sucrose release was tested using a sucrose kit. 30 μL of supernatant and 2 mL of hydrolysed solution were thoroughly mixed and subsequently heated in a water bath at 100 °C for 8 min. Sucrose was hydrolysed in an acidic hydrolysate to produce glucose and fructose. Fructose further converts into 5-hydroxymethylfurfural in an acidic hydrolysate, which possesses a maximum absorption wavelength of 290 nm. Therefore, by monitoring the absorbance of the solution at 290 nm together with the sucrose calibration curve, the quantitative detection of sucrose was achieved.

## S. cerevisiae cultivation

*S. cerevisiae* (Mingzhoubio, China) were suspended in 0.5 mL of YM medium (21 g/L). The resuspension (200 μL) was spread on the prepared YM solid medium and incubated at 28 °C for 48 h. The colonies grown from the solid medium were picked and placed in the YM liquid medium, which was shaken for 24 h at 28 °C.

## Construction of a two-species community and its pH oscillatory environment

The two-species community contains pH-responsive artificial cells containing sucrose (species A) and *S. cerevisiae* (species B). Species B cultured for 24 h was centrifuged (800 × g, 8 min) in a 4 mL centrifuge tube to remove the medium. The precipitate was resuspended in a mixture of gadobutrol solution, followed by centrifugation (800 × g, 8 min) to remove the supernatant. The above procedure was repeated 3 times to remove the residual medium. Species A was prepared by the emulsion method.

Species B was mixed with species A (containing 300 mM sucrose, 50 mM Na₂CO₃, 0.05 g/mL PEG, 0.01 mol/L pH-responsive molecule) in gadobutrol solution (400 mM, 500 μL). The number ratio of species B and species A is 61.2%:38.8%. Melittin was added to the solution at a final concentration of 1 μg/mL. The pH of the environment of the community was monitored in real time using a precise micro pH meter (Mettler Toledo, Switzerland). Sucrose molecules were first released from species A to be consumed by species B to produce $CO_2$. The production of $CO_2$ led to a decrease of solution pH, which further caused the slow releasing rate of sucrose. The dissipation of $CO_2$ into

the air became the dominant process, which shifted the balance in Eq. 5 to the left, consequently increased the solution pH.

$$CO_2 + H_2O \rightleftharpoons HCO_3^- + H^+ \tag{5}$$

## Construction of the three-species community and their interactions

The three-species community contains pH-responsive artificial cells containing sucrose and G6P (species A′), *S. cerevisiae* (species B) and artificial cells containing NAD⁺ and G6PDH (species C). Species B was mixed with species A′ (containing 150 mM sucrose, 50 mM Na₂CO₃, 150 mM lactose, 0.05 g/mL PEG, 10 mmol/L pH-responsive molecule, 10 mM G6P) and species C (containing 300 mM lactose, 50 mM Na₂CO₃, 10 mM NAD⁺, 10 μg/mL G6PDH) in gadobutrol solution (400 mM, 500 μL). The ratio of species A′, species B and species C was 41.2%:29.9%:29.9%. Melittin was added to the solution at a final concentration of 1 μg/mL. The fluorescence change of NAD⁺-GUVs was monitored by a fluorescence spectrometer (PerkinElmer, USA) and fluorescence microscope.

## Construction of a spatially coded three-species community using the Magneto-Archimedes effect

A spatially coded three-species community was constructed using the Magneto-Archimedes effect in a homemade device. The device contained two coverslips separated by a hollow Teflon spacer (inner square of 1.0 cm × 1.0 cm), which sat on a NdFeB magnet. An SS mesh was placed on the bottom coverslip to produce an inhomogeneous magnetic field distribution inside each well. The device was sealed with vacuum lubricating grease. The Magneto-Archimedes effect was determined by Eq. 6[48]:

$$U(r) = -2\pi R^3 \mu_0 \frac{\chi_G - \chi_S}{\chi_G + 2\chi_S + 3} |H(r)|^2 \tag{6}$$

where U (r) is the static magnetic potential energy of the GUV with radius R at position r, $\mu_0$ is the magnetic permeability of vacuum, $\chi_G$ and $\chi_S$ are the magnetic susceptibilities of the GUV and solution, respectively, and H (r) is the magnetic field at position r. If $\chi_G$ is less than $\chi_S$, the GUV in the solution tends to move towards the region with the weakest magnetic field to lower U (r). By varying the addition order of pH-responsive artificial cells containing sucrose and G6P (species A′), *S. cerevisiae* (species B) and artificial cells containing NAD⁺ and G6PDH (species C), A′CB, CA′B and CBA′ were obtained. For dye penetration experiments, species A″ (containing G6P, fluorescein and sucrose), species B and species C were coded to obtain A″CB and CBA′′. To prevent GUVs from breaking during the experiment, 300 μL of DPPC ethanol-water solution (0.10 mg/mL, ethanol: water = 40%:60%, v/v) was added to coverslips, followed by heating at 60 °C for 10 min and washing 3 times with 400 mM gadobutrol solution. All species were dispersed in gadobutrol solutions (400 mM, 500 μL). Melittin was added to the solution at a final concentration of 1 μg/mL. The fluorescence change of species C was monitored by laser scanning confocal microscopy.

## Reporting summary

Further information on research design is available in the Nature Portfolio Reporting Summary linked to this article.

## Data availability

Data supporting the findings of this work are available within the paper and its Supplementary Information files. A reporting summary for this Article is available as a Supplementary Information file. Source data are provided with this paper.

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

## Acknowledgements

This work was supported by the National Natural Science Foundation of China (Grant No. 22174031, 21929401, 22111540252, and 22374033 to X.H.), Natural Science Foundation of Heilongjiang Province grant ZD2022B001 to X.H., Heilongjiang Touyan Team grant HITTY-20190034 to X.H., Fundamental Research Funds for the Central Universities grant HIT.OCEF.2021026 to X.H.

## Author contributions

Conceptualisation: X.H., L.T. and S.L.; Methodology: S.L., Y.Z., L.T. and X.H.; Investigation: S.L., Y.Z., S.W., X.Z., and B.Y.; Visualisation: S.L., Y.Z., S.W., X.Z., and B.Y.; Funding acquisition: X.H.; Project administration: X.H.; Supervision: X.H. and L.T.; Writing – original draft: S.L. and L.T.; Writing – review & editing: S.L. and X.H.

## Competing interests

The authors declare no competing interests.
