## [Peer Review File · Nature Communications]

Regulation of species metabolism in synthetic community systems by environmental pH oscillationsReviewers' Comments:

Reviewer #1:

Remarks to the Author:

Li et al produce what they refer to as artificial cells that can interact with yeast and another type of artificial cell. Importantly, feedback cycles were possible between the artificial cells and yeast. The artificial cell was simply a phospholipid vesicle containing "pH responsive molecules" and sucrose. The membrane possessed melittin pores. At low pH, a gel formed, whereas at higher pH, the solution was less viscous. Released sucrose (at high pH) was consumed by the yeast, leading to a decrease in pH through the release of CO₂ (carbonate). The low pH blocked the release of more sucrose. Over time, through equilibration with the atmosphere, the pH increased again, leading to more sucrose being released and the commencement of another cycle.

The cyclic nature of the interaction of the artificial cells and the yeast was to me the advance in the work. However, I don't consider what was built an artificial cell. It's much too simplistic. There's not even nucleic acid involved. Nothing is genetically encoded.

It is nice that a pH responsive system was built, but that's not very novel. There are many publications on pH responsive hydrogels and coacervates.

It also was not clear from the main text what molecules were used. I don't think the pH responsive molecule was ever named, although the structure was given in Fig. S1.

Similarly, the properties of melittin were never explained.

The need for gadobutrol was not explained (nor what gadobutrol was).

Some of the information should probably be moved to the supplemental section, including Fig. 2b and Fig. 3.

The three species system consists of an extra "artificial cell" that receives released glucose 6-phosphate and then with an enzyme (which wasn't explained) reduced NAD⁺ to NADH, but this extra artificial cell did not seem to add anything to the story. What did we learn from this?

The spatial control of the location of the participating artificial cells is certainly "cool" but the data did not seem to tell us anything more than we would have already guessed.

For me, the "two species" system is interesting and seems like a nice platform from which more could be built, but I also do not think that this is a big advancement from previous work.

I would strongly suggest to the authors during revisions to more clearly and thoroughly explain how the experiments were run, the purpose of the different molecules that were used, and what can be learned from three species and spatial control experiments.

Reviewer #2:

Remarks to the Author:

This report describes the use of artificial cells and yeast cells to develop an oscillating system based on a feedback mechanism. The pH of the medium oscillates around 6.4, which is thought to be the result of the gelation-dependent release of sucrose and the subsequent production of CO₂. A third population is added to the system, in which the positioning of the three populations has an effect on the overall activity. The construction of artificial cell populations with interactive mechanisms is of interest to the synthetic cell community. The combination with yeast cells is not often observed. Still

there are major issues with this manuscript that need to be addressed.

First of all, I don't understand the oscillation between pH 6.5 and 6.4. The viscosity change and subsequent difference in release of sucrose is very small in this window, and since the reset (release of CO₂ from the system) is the slowest process, there is no reason why not a steady state should be obtained in which, due to differences in rates, the system finds a pH in which production and release are in balance. Obtaining oscillations is notoriously difficult as the different rates have to be matched very well. It is highly surprising that under all measured conditions oscillations take place. Why do you still see oscillations at pH 7, when there is no gelation effect? The authors should therefore study the individual rates of the different steps in more detail in order to better understand how this system really works.

Furthermore, why does the pH increase also take longer in subsequent cycles? This is a pure physical process (release from CO₂ to the air) and should not be affected by the concentration of sucrose.

Secondly, the pH of a CO₂ saturated system should be more close to 4 at atmospheric pressure. It is therefore not logical that the pH increases that effectively.

The authors did not perform a control reaction in which they use only yeast cells and vary the sucrose concentration in the window that is achievable by the release from artificial cell A.

I find it difficult to see how the closely packed cell populations could prevent diffusion of small molecules or protons. The authors should study this in more detail with e.g. model dye components.

Reviewer #3:

Remarks to the Author:

Han et al. tried to construct unique artificial cells and create cell communities consisting of artificial cells and natural cells. The core technique to accomplish this research is the use of pH-sensitive molecule that phase-shift between fluid and gel phases, and encapsulate it inside giant vesicles. By changing the surrounding pH, the artificial cells release sugar into the environment and the neighbor yeasts consume the sugar and release CO₂ into the environment. As a consequence, the oscillation of pH change was generated in the society. The authors developed the society more complicated by increasing the species as 2 types of artificial cells and yeasts. Finally, the authors demonstrate the controlled communication between the species by giving spatial constraints.

The concept of an artificial community with artificial cells and natural cells is previously proposed by Mansy et al. (2014) and Stano et al. (2018). But this work is further advanced with fine-tuning. This would be a noteworthy paper in the field of artificial cell research which is rapidly developing nowadays.

The data are clear and analyzed well. Figures are well arranged to easily understand the results. But, unfortunately, the description in the Discussion is poor. The authors just repeat the explanation of what they did but mentioned nothing further. The authors should describe, at least, how the artificial system achieved in this work will benefit the study of biological phenomena, or how does it useful for controlling the biological system.

Therefore, before accepting the paper, the reviewer believes that the discussion part should be enriched so that the readers can understand what value this work has for life science as a whole.

The following are questions and minor revisions.

Page 6, Fig. 1, Is it possible to know the rate of release of the sucrose from the vesicles at pH 6.6?

Page 9, Fig. 2, Did the yeast proliferate during the cycle turning?

Page 9, Fig. 2, Revise the head title of Fig. 2.

Page 11, line 175, "7.2)" -> "7.2"

Page 13, line 217, "Fig. 3C". This might be "Fig. 4c".

Page 13, line 227, "Fig. 3d". This might be "Fig. 4c".

G. Rampioni et al. (Chemical Communications 2018) also reported pioneering work in this research line. This should be cited in the text.

Response to the reviewers' comments

For the sake of clarity, the comments of the reviewer have been collated in black, and our response to each comment appears in blue. All the changes to the manuscript are highlighted in red.

Reviewer #1 (Remarks to the Author):

Li et al produce what they refer to as artificial cells that can interact with yeast and another type of artificial cell. Importantly, feedback cycles were possible between the artificial cells and yeast. The artificial cell was simply a phospholipid vesicle containing "pH responsive molecules" and sucrose. The membrane possessed melittin pores. At low pH, a gel formed, whereas at higher pH, the solution was less viscous. Released sucrose (at high pH) was consumed by the yeast, leading to a decrease in pH through the release of CO₂ (carbonate). The low pH blocked the release of more sucrose.

Over time, through equilibration with the atmosphere, the pH increased again, leading to more sucrose being released and the commencement of another cycle.

1) The cyclic nature of the interaction of the artificial cells and the yeast was to me the advance in the work. However, I don't consider what was built an artificial cell. It's much too simplistic. There's not even nucleic acid involved. Nothing is genetically encoded.

Thank the reviewer for the comment. The artificial cells refer to the structures mimicking partial/whole cell structure and functions. The cell structures include membrane, organelles, cytosol etc. The cell functions are metabolism, division, mass transport across the membrane, etc. The phospholipid vesicles containing non-biological components or enzymes were called artificial cells (Nat. Commun. 2014, 5, 5305; PNAS 2019, 116, 16711; Nat. Chem. Biol. 2018, 14, 86) in this field. We call our phospholipid vesicle structures as artificial cells because they mimicked the cell function of mass transport across the membrane, as well as the cell responsive function to environmental stimulus. We added the description of artificial cells in the introduction as below in page 3.

[...] The understanding of these complex dynamic behaviors will help us to decode the operating principles of biological system that support and maintain life, and also pave the way to advance future microscale technologies exhibiting key features of living systems.⁶⁻¹¹ **Artificial cells are structures mimicking partial/whole cell structure and functions prepared using natural or synthetic materials,¹²⁻¹⁴ which help to understand the working mechanism of cells.¹⁵⁻²² [...]**

2) It is nice that a pH responsive system was built, but that's not very novel. There are many publications on pH responsive hydrogels and coacervates.

Thank the reviewer for the comment. Most of the current pH responsive systems were regulated by the environment, which exhibited one-way response manner. To the best of our knowledge, it is the first time to encapsulate pH responsive hydrogel in phospholipid vesicles, which communicate the environment in a bidirectional response manner via the interaction with yeasts, consequently to generate pH oscillation of solution.

3) It also was not clear from the main text what molecules were used. I don't think the pH responsive molecule was ever named, although the structure was given in Fig. S1.

Thank the reviewer for the comment. We have named the molecules and modified the text as below in page 21.

[...] The compound was dried overnight at 50 °C under reduced pressure. The compound (Supplementary Fig .1) was pH-responsive molecules ((S)-4-((1-(dodecylamino)-3-methyl-1-oxobutan-2-yl)amino)-4-oxobutanoic acid) with yield of 95%.

4) Similarly, the properties of melittin were never explained.

Thank the reviewer for the comment. We have added a description of the properties of melittin in the main text as below in page 4.

[...] The pH-responsive molecules changed from fluid phase at high pH value (≥ 7) into gel phase at low pH value (≤ 6) due to the hydrogen bonds (Fig. 1a and b). Melittin is a 26-amino-acid α -helical peptide, which forms pores on the lipid bilayer membrane for substance exchange between internal and external of lipid vesicles.^{35,36} Here, the melittin nanopores on the bilayer membrane allowed protons exchange between inner artificial cells and external environments. [...]

5) The need for gadobutrol was not explained (nor what gadobutrol was).

Thank the reviewer for the comment. We have added a description of gadobutrol and its necessity as below in page 15.

[...] Spatial coded three-species-communities were constructed inside each well of a steel mesh (Supplementary Fig .16) under a magnetic field using Magneto-Archimedes principle with a homemade device^{37,38} (Supplementary Fig .17). Magneto-Archimedes effect was determined by equation (2)³⁹:

$$U(r) = -2\pi R^3 \mu_0 \frac{\chi_G \chi_S}{\chi_G + 2\chi_S + 3} |H(r)|^2 \quad (2)$$

where $U(r)$ is the static magnetic potential energy of GUVs with radius R at r position, μ_0 is the magnetic permeability of vacuum, χ_G and χ_S are the magnetic susceptibility of GUV and solution respectively, $H(r)$ is the magnetic field at r position. If χ_G is less than χ_S , the GUV in the solution tends to move towards the region with the weakest magnetic field to lower $U(r)$. Gadobutrol is a paramagnetic contrast agent used in magnetic resonance imaging.⁴⁰ The addition of gadobutrol aimed to increase χ_S . Three patterns (CA'B, CBA', A'CB) of three-species-communities were obtained by varying addition order of species A', species B, and species C (Supplementary Fig .18). [...]

6) Some of the information should probably be moved to the supplemental section, including Fig. 2b and Fig. 3.

Thank the reviewer for the comment. We have moved original Fig. 2b and Fig. 3 to the supplemental section as Supplementary Fig .5, Supplementary Fig .9 and Supplementary Fig .12, respectively.

7) The three species system consists of an extra "artificial cell" that receives released glucose 6-phosphate and then with an enzyme (which wasn't explained) reduced NAD+ to NADH, but this extra artificial cell did not seem to add anything to the story. What did we learn from this?

Thank the reviewer for the comment. We supplemented the descriptions of NADH and G6PDH in the main text. Glucose-6-phosphate dehydrogenase (G6PDH) is an important enzyme involved in glycolysis pathway. G6PDH inside species C can convert glucose-6-phosphate and NAD^+ to 6-

phosphate gluconate lactone and NADH respectively. Therefore, species C is an artificial cell for metabolic mimicry. The introduction of species C increased the complexity of communities from two-species-community to three-species-community. The biochemical reaction in species C was adjusted by the communications between species A' and B. The species A', B, C and environment were an integral system, among which these elements activities were influenced by others. We established a three-species-community with complex internal dynamic interactions, which helps to understand the behavior of complex community. We modified the main text as below in pages 12 and 13.

[...] The controlled release of G6P from species A' was expected to trigger the stepped generation of NADH inside species C (Fig. 3a). **Glucose-6-phosphate dehydrogenase (G6PDH) is an important enzyme involved in glycolysis pathway. Upon receiving G6P, the G6PDH inside species C converted G6P and NAD⁺ to 6-phosphate gluconate lactone and NADH, respectively. NADH is important biomolecule involved in many metabolic pathways. Species C is an artificial cell capable of metabolism mimicry.**

[...] No NADH was generated inside species C with the absence of G6PDH in species C (Fig. 3e, blue curve), NAD⁺ in species C (Fig. 3e, black curve) or G6P in species A (Fig. 3e, red curve), which confirmed the feedback between species A' and species B regulated the internal 'metabolism' in species C. **The species A', B, C and environment were an integral system, among which these elements activities were influenced by others. We established a three-species-community with complex internal dynamic interactions, which helps to understand the behavior of complex community.**

8) The spatial control of the location of the participating artificial cells is certainly "cool" but the data did not seem to tell us anything more than we would have already guessed.

Thank the reviewer for the comment. The spatial distribution of cells plays a crucial role for cell function to the community. Here, we have demonstrated for the first time the importance of spatial location distribution in synthetic communities using a three-species-community constructed using magnetic fields. We found the spatial distribution of species in a community influenced their internal interactions, which helps to understand the structure of communities. In addition, we performed dye diffusion experiments in different spatial coded communities using fluorescein as a model dye to observe the inhibitory effect of tightly packed species on the diffusion of small molecules more intuitively. We modified the text as below in pages 16,18 and 24.

[...] The intensity of community A'CB (Fig. 4i) continuously increased, because the release of G6P from species A' at relative high pH (Fig. 4g) diffused into the adjacent species C to generate NADH. **In order to observe the inhibitory effect of tightly packed species on the diffusion of small molecules more intuitively, we prepared a species A'' encapsulating model dye molecule of fluorescein (Mw=332.3 g/mol), which has green fluorescence and molecular weight similar to sucrose (Mw=342.3 g/mol) (Supplementary Fig .21a). Two three-species communities were constructed by arranging species A'', species B (Supplementary Fig .21b), and species C (labeled with TR DHPE, Supplementary Fig .21c) in the order of A''CB (Supplementary Fig .21d, e, and f) and CBA'' (Supplementary Fig .21h, i, and j). No green fluorescence was observed in the species B region in A''CB community within 270 minutes (Supplementary Fig .21g) due to the close pack of species C. Similarly, no green fluorescence was observed in the species C region due to the close pack of species B (Supplementary Fig .21k). All abovementioned results demonstrated that the spatial distribution of species in the community did exhibit dramatic impact on the behavior of community system.**

[...] The location of species in the spatial coded three-species-community system is confirmed to have dramatic impact on the interaction among species and between community and its environment. **The species in natural communities often exist in spatial orders. The influence study of spatial**

distribution on signal transmission among species in spatial coded synthetic communities helps to understand the structure and function of communities.

[...] By varying the addition order of pH-responsive artificial cells containing sucrose and G6P (species A'), yeasts (species B) and artificial cells containing NAD⁺ and G6PDH (species C), A''CB, CA'B and CBA'' were obtained. For dye penetration experiments, species A'' (containing G6P, fluorescein and sucrose), species B and species C were coded to obtain A''CB and CBA''. To prevent GUVs from breaking during the experiment, 300 μ L of DPPC ethanol-water solution (0.10 mg/mL, ethanol: water = 40 %:60 %, v/v) was added to coverslips, followed by heating at 60 $^{\circ}$ C for 10 min and washing 3 times with 400 mM gadobutrol solution. [...]

Supplementary Fig .21. (a) Species A'' encapsulating fluorescein and pH-responsive molecules. (b) Yeast (species B). (c) Species C labelled with TR DHPE. Schematic diagram of A''CB community with top view (d) and side view (e). (f) Merged image of laser scanning confocal images of A''CB community taken with red and green channels, and white field. (g) Laser scanning confocal microscopy images of A''CB community with green channel over time. Schematic diagram of CBA'' community with top view (h) and side view (i). (j) Merged image of laser scanning confocal images of CBA'' community taken with red and green channels, and white field. (k) Laser scanning confocal

microscopy images of CBA'' community with green channel over time. The scale bars were 20 μm in a, b and c, and 100 μm in f, g, j, and k.

9) For me, the "two species" system is interesting and seems like a nice platform from which more could be built, but I also do not think that this is a big advancement from previous work.

Thank the reviewer for the comment. The advancement of our work is the involvement of environment (solution) in the community system, rather than only taking into accounts of species. More importantly, the solution pH oscillation (environment variation) caused by two species influenced the 'metabolic pathway' of the third species. Therefore, we established a community possessing complex dynamic interaction network, which provide a way for investigating complicated inter network reaction among micro-ecosystems, and laid the foundation for building more complex systems in the future to perform higher-order functions. We modified the main text as below in pages 12 and 18.

[...] the balance of CO_2 dissipation rate and CO_2 production rate caused lower steady median pH value, bigger amplitude and shorter period of oscillation at higher percentage of species A.

The construction of a two-species-community oscillation system provides the possibility of constructing complex dynamic response systems using simple components. Although two-species-community is a useful platform for studying inter species interactions, natural communities typically consist of more than two species to maintain community stability. In the following content, a three-species community was constructed to investigated the interactions among species and environment.

[...] The location of species in the spatial coded three-species-community system is confirmed to have dramatic impact on the interaction among species and between community and its environment. The species in natural communities often exist in spatial orders. The influence study of spatial distribution on signal transmission among species in spatial coded synthetic communities helps to understand the structure and function of communities.

10) I would strongly suggest to the authors during revisions to more clearly and thoroughly explain how the experiments were run, the purpose of the different molecules that were used, and what can be learned from three species and spatial control experiments.

Thank the reviewer for the comment. We have added more experimental details, experimental design, and the significance of three species and spatial control experiments as below in pages 4, 12, 13, 15 and 18.

Page 4

[...] The pH-responsive molecules changed from fluid phase at high pH value (≥ 7) into gel phase at low pH value (≤ 6) due to the hydrogen bonds (Fig. 1a and b). Melittin is a 26-amino-acid α -helical peptide, which forms pores on the lipid bilayer membrane for substance exchange between internal and external of lipid vesicles.^{35,36} Here, the melittin nanopores on the bilayer membrane allowed protons exchange between inner artificial cells and external environments. [...]

Page 12

[...] The controlled release of G6P from species A' was expected to trigger the stepped generation of NADH inside species C (Fig. 3a). Glucose-6-phosphate dehydrogenase (G6PDH) is an important enzyme involved in glycolysis pathway. Upon receiving G6P, the G6PDH inside species C converted G6P and NAD^+ to 6-phosphate gluconate lactone and NADH, respectively. NADH is important biomolecule to provide reductive force in many metabolic pathways. Species C is an

artificial cell capable of metabolism mimicry.

Page 13

[...] No NADH was generated inside species C with the absence of G6PDH in species C (Fig. 3e, blue curve), NAD⁺ in species C (Fig. 3e, black curve) or G6P in species A (Fig. 3e, red curve), which confirmed the feedback between species A' and species B regulated the internal 'metabolism' in species C. The species A', B, C and environment were an integral system, among which these elements activities were influenced by others. We established a three-species-community with complex internal dynamic interactions, which helps to understand the behavior of complex community.

Page 15

[...] Spatial coded three-species-communities were constructed inside each well of a steel mesh (Supplementary Fig .16) under a magnetic field using Magneto-Archimedes principle with a home-made device^{37,38} (Supplementary Fig .17). Magneto-Archimedes effect was determined by equation (2)³⁹:

$$U(r)=-2\pi R^3\mu_0\frac{\chi_G-\chi_S}{\chi_G+2\chi_S+3}|H(r)|^2 \quad (2)$$

where U (r) is the static magnetic potential energy of GUVs with radius R at r position, μ_0 is the magnetic permeability of vacuum, χ_G and χ_S are the magnetic susceptibility of GUV and solution respectively, H (r) is the magnetic field at r position. If χ_G is less than χ_S , the GUV in the solution tends to move towards the region with the weakest magnetic field to lower U (r). Gadobutrol is a paramagnetic contrast agent used in magnetic resonance imaging.⁴⁰ The addition of gadobutrol aimed to increase χ_S . Three patterns (CA'B, CBA', A'CB) of three-species-communities were obtained by varying addition order of species A', species B, and species C (Supplementary Fig .18). [...]

Page 18

[...] The location of species in the spatial coded three-species-community system is confirmed to have dramatic impact on the interaction among species and between community and its environment. The species in natural communities often exist in spatial orders. The influence study of spatial distribution on signal transmission among species in spatial coded synthetic communities helps to understand the structure and function of communities.

Reviewer #2 (Remarks to the Author):

This report describes the use of artificial cells and yeast cells to develop an oscillating system based on a feedback mechanism. The pH of the medium oscillates around 6.4, which is thought to be the result of the gelation-dependent release of sucrose and the subsequent production of CO₂. A third population is added to the system, in which the positioning of the three populations has an effect on the overall activity. The construction of artificial cell populations with interactive mechanisms is of interest to the synthetic cell community. The combination with yeast cells is not often observed. Still there are major issues with this manuscript that need to be addressed.

1) First of all, I don't understand the oscillation between pH 6.5 and 6.4. The viscosity change and subsequent difference in release of sucrose is very small in this window, and since the reset (release of CO₂ from the system) is the slowest process, there is no reason why not a steady state should be obtained in which, due to differences in rates, the system finds a pH in which production and release are in balance. Obtaining oscillations is notoriously difficult as the different rates have to be matched very well. It is highly surprising that under all measured conditions oscillations take place. Why do you still see oscillations at pH 7, when there is no gelation effect? The authors should therefore study the individual rates of the different steps in more detail in order to better understand how this system really works.

Thank the reviewer for the comment. To better understand how this oscillating system works, we studied the effect of pH on the leakage of sucrose from artificial cells by measuring the concentrations of sucrose released to the external solution at different pH (Fig. R1a). A new parameter 'relative diffusion coefficient' was extracted to describe the leaking ability of sucrose driven by the concentration difference of the inner and outer membrane, showing that there was a gelation effect at a wide range of pH from 6 to 7 (Fig. R1b). Although the diffusion difference was very small, the coupled nonlinear processes may amplify this difference, leading to continuous oscillations.

Fig. R1. a, Time-dependent plot of sucrose concentrations released from species A (containing 400 mM sucrose) in the different environment solution pH. **b,** Plot of relative diffusion coefficient of sucrose at different environment solution pH, derived from (a).

The oscillation range of T_1 period in Fig. 2b is 6.63 to 6.42, which is closer to 6.6 and 6.4. The pH-oscillating system was the result of coupled multiple nonlinear processes, which constructed a closed feedback loop, where the negative feedback motif from H⁺ to sucrose (Fig. R2a) played an important role in oscillation generation. Fig. R2a is the scheme of non-linear feedback network in the oscillation system. In order to further investigate the contribution of the diffusion difference to the oscillating system, we simplified the complex system to a simple oscillation model based on the binary function (see below 'Oscillation computational model I' section), where the relative diffusion coefficient of sucrose was set 3×10^{-13} cm²/s in the case of pH greater than 6.6, while it was set 0 in

the case of pH less than 6.6. The setting value of $3 \times 10^{-13} \text{ cm}^2/\text{s}$ was the diffusion difference between pH 6.6 and 6.4. We found that oscillation can be generated under this condition (Fig. R2b, black curve), but when its difference was reduced to $0.8 \times 10^{-13} \text{ cm}^2/\text{s}$, the system would reach a steady state (Fig. R2b, red curve), indicating that the pH-responsive diffusion differences could be amplified to generate pH-responsive oscillations through multiple nonlinear processes.

In addition, the appearance of oscillations was the clever coupling of different parameters during the process, where the escape of CO_2 was also a critical part. Open system was another necessary condition for continuous oscillations. But it is hard to calculate the certain value of mass transfer coefficient of CO_2 , because there were multiple complicate influencing factors in this process. However, from the simulation, we guess that the mass transfer coefficient of CO_2 can be coupled with other parameters in a relatively wide range ($0.01 \text{ cm/s} \sim 1 \times 10^{-10} \text{ cm/s}$) to obtain oscillations (Fig. R2c).

Fig. R2. a, The scheme of non-linear feedback network in the oscillation system. **b**, Time-dependent plot of pH under relative diffusion coefficient of $3 \times 10^{-13} \text{ cm}^2/\text{s}$ (black curve) and $0.8 \times 10^{-13} \text{ cm}^2/\text{s}$ (red curve) obtained by oscillation computational model I. The relative diffusion coefficient of sucrose was set as $3 \times 10^{-13} \text{ cm}^2/\text{s}$ and $0.8 \times 10^{-13} \text{ cm}^2/\text{s}$ in the case of pH greater than 6.6, while it was set as 0 in the case of pH less than 6.6. K_{CO_2} was set as 0.01 cm/s. **c**, Time-dependent plot of pH under different mass transfer coefficient in oscillation computational model I. K_{CO_2} was set to be 0.01 and $1 \times 10^{-10} \text{ cm/s}$, respectively. The relative diffusion coefficient of sucrose was set as $1 \times 10^{-12} \text{ cm}^2/\text{s}$.

The reason for oscillation at pH 7 is that gelation effect still exists around pH 7. The viscosities of internal artificial cell are $5.09 \times 10^{-3} \text{ mPa}\cdot\text{s}$ at pH 7 and $1.47 \times 10^{-3} \text{ mPa}\cdot\text{s}$ at pH 8. (Supplementary Fig. 2) Considering the process of sucrose releasing from the artificial cells was influenced by multiple factors, such as the membrane permeability besides viscosity, we introduce a parameter 'relative diffusion coefficient' of sucrose to qualitatively study the effect of pH on sucrose leakage (Fig. R1b). A diffusion difference of sucrose leakage around pH 7 was observed (Fig. R1b). Through the oscillation model I, in which the binary function was performed as relative diffusion coefficient of sucrose was set as $1.5 \times 10^{-13} \text{ cm}^2/\text{s}$ in the case of pH greater than 7, while it was set as 0 in the case of pH less than 7. The simulation confirmed the oscillation could be generated under this condition (Fig. R3).

Fig. R3 Time-dependent plot of pH under relative diffusion coefficient of $1.5 \times 10^{-13} \text{ cm}^2/\text{s}$

The introduction of relative diffusion coefficient

The new parameter 'relative diffusion coefficient' was introduced to describe the rate of sucrose releasing out of pH-responsive artificial cells. We simplified the process of molecule diffusion to one driven by the concentration difference of the inner and outer membrane. The relative diffusion coefficient was determined by Fick's equation:

$$\frac{\partial[\text{sucrose}]}{\partial t} = D'_{\text{sucrose}} \left(\frac{\partial^2[\text{sucrose}]}{\partial x^2} + \frac{\partial^2[\text{sucrose}]}{\partial y^2} + \frac{\partial^2[\text{sucrose}]}{\partial z^2} \right) \quad (1)$$

The finite explicit approach was used to simplify equation (1):

$$\frac{[\text{sucrose}]_{\text{in}(t+1)} - [\text{sucrose}]_{\text{in}(t)}}{\Delta T} = -6D'_{\text{sucrose}} \frac{[\text{sucrose}]_{\text{in}(t)} - [\text{sucrose}]_{\text{out}(t)}}{d^2} \quad (2)$$

where $[\text{sucrose}]_{\text{in}}$ was the concentration of sucrose inside the artificial cells; $[\text{sucrose}]_{\text{out}}$ was the concentration of sucrose outside the artificial cells; D'_{sucrose} denoted relative diffusion coefficient of sucrose; ΔT was the time interval, and d represented the diffusion distance determined by the concentration of artificial cells. Here d was set as $6 \mu\text{m}$ with artificial cells concentration of 3.7×10^8 cells/mL and cells size of $5 \mu\text{m}$.

The Boltzmann model was used to fit the relationship between the relative diffusion coefficient of sucrose and the environment solution pH:

$$D'_{\text{sucrose}} = \left(2.72 - \frac{2.53}{1 + \exp\left(\frac{\text{pH} - 6.65}{0.29}\right)} \right) \times 10^{-12} \quad (3)$$

Oscillation computational model I

In the system, sucrose-entrapped artificial cells and yeasts were assumed to be dispersed homogenously. At high environment solution pH, the membrane of artificial cells showed high permeability to sucrose, which were consumed by yeasts to generate CO_2 , leading to the decrease of environment solution pH. In this case, sucrose stopped releasing. As CO_2 escaped into the air, the environment solution pH has rebounded apparently, triggering next round of pH oscillations.

The whole reactions can be described as below, based on the reaction rate equations:

$$\frac{\partial[\text{sucrose}]}{\partial t} = -k_1[\text{sucrose}] \quad (4)$$

$$\frac{\partial[\text{CO}_2]}{\partial t} = 12k_1[\text{sucrose}] - k_2[\text{CO}_2] + k_3[\text{HCO}_3^-][\text{H}^+] \quad (5)$$

$$\frac{\partial[\text{H}^+]}{\partial t} = k_2[\text{CO}_2] - k_3[\text{HCO}_3^-][\text{H}^+] \quad (6)$$

where k_1 was set as 15.3 min^{-1} (Open J. Phys. Chem., 2014, 4 (1), 26-31), k_2 was set as 0.037 s^{-1} (Mar. Chem., 2006, 100 (1-2), 53-65), and k_3 was set as $0.076 \mu\text{M}^{-1} \text{ s}^{-1}$ (1982, 27 (5), 849-855).

The leakage of sucrose from artificial cells was simplified to be a binary function, where the relative diffusion coefficient of sucrose was set as $3 \times 10^{-13} \text{ cm}^2/\text{s}$ in the case of pH greater than 6.6, while it was set as 0 in the case of pH less than 6.6. The concentration of sucrose, CO_2 , and H^+ in the solution were assumed to be uniform, because their diffusion ($D(\text{sucrose}) = 1.86 \times 10^{-6} \text{ cm}^2/\text{s}$ (J. Phys. Chem. A, 2003, 107 (6), 936-943); $D(\text{CO}_2) = 1.85 \times 10^{-5} \text{ cm}^2/\text{s}$ (J. Agric. Food Chem., 2003, 51 (26), 7560-7563)) were too fast, compared to the process of sucrose leaking out of artificial cells.

The escape of CO_2 into the air can be described based on two-film theory:

$$N_{\text{CO}_2} = K_{\text{CO}_2}([\text{CO}_2]_{\text{in}} - [\text{CO}_2]_{\text{out}}) \quad (7)$$

Here, N_{CO_2} was represented as mass transfer flux; K_{CO_2} was represented as mass transfer coefficient, and set as 0.01, which was just a hypothetical value; $[\text{CO}_2]_{\text{in}}$ was the concentration of CO_2 in the solution; $[\text{CO}_2]_{\text{out}}$ was the concentration of CO_2 in the air.

2) Furthermore, why does the pH increase also take longer in subsequent cycles? This is a pure physical process (release from CO_2 to the air) and should not be affected by the concentration of sucrose.

Thank the reviewer for the comment. The extension of the oscillation period was attributed to the reduction of the sucrose concentration difference between the inner and outer membrane. Due to the continuous release of sucrose, the concentration difference decreased in the subsequent cycles, resulting in the slower production of H^+ . In the pH-rising phase of oscillation cycle, CO_2 was continuously produced in the system. Therefore, it wasn't a pure physical process of CO_2 releasing to the air.

To support this view, we constructed a simple model that abstracted the oscillating system into a nonlinear feedback network, where the introduction of k_1' simulated the self-producing ability of sucrose, which came from the diffusion ability of sucrose from artificial cells. (Fig. R4a) The rise of internal concentration of sucrose or density of artificial cells would be performed as the increase of k_1' . By setting larger value of k_1' in the simulation, we found that the amplitude would increase while the period would decrease, which echoed the experimental results, proving the feasibility of the simple model (Fig. R4b). In the improvement of the model, we set k_1' as a parameter that gradually decreased over time, due to the continuous release of sucrose from the artificial cells in the subsequent cycles. The oscillation periods were observed gradually longer (Fig. R4c), confirming that the extension of the oscillation cycle was the result of the reduction of the sucrose concentration difference. Meanwhile the pH increase took longer in the subsequent cycles.

Oscillation computational model II

In order to investigate the influencing factors of oscillation period, we developed another oscillation computational model based on the negative feedback loop. The whole reactions could be described as below:

$$\frac{\partial[A]}{\partial t} = k_1' - k_2'[A][B] - k_4'[A][C] \quad (8)$$

$$\frac{\partial[B]}{\partial t} = k_2'[A][B] - k_3'[B][C] - k_5'[B] \quad (9)$$

$$\frac{\partial[C]}{\partial t} = k_3'[B][C] - k_4'[A][C] \quad (10)$$

where k_1' was set as 1; k_2' was set as 1; k_3' was set as 1; k_4' was set as 0.01; k_5' was set as 0.1.

Fig. R4. a, The scheme of non-linear feedback network in the oscillation computational model II. b, Time-dependent plot of the negative number of the logarithm of 'C' concentrations under different value of k_1' in oscillation computational model II. The value of k_1' was set as 1 and 10 respectively. c, Time-dependent plot of the negative number of the logarithm of 'C' concentrations under different value of k_1' in oscillation computational model II. The value of k_1' was set as 1 and 10 respectively.

c, Time-dependent plot of the negative number of the logarithm of 'C' concentrations in oscillation computational model II, where $k_1' = 2 - 0.02t$; $k_2' = 1$; $k_3' = 0.7$; $k_4' = 0.01$; and $k_5' = 0.1$.

We modified the main text as below in pages 8.

[...] It can be explained by the consumption of sucrose inside species A, which resulted the difference of the concentration of solution components at each equilibrium point in subsequent cycles. The decrease of the sucrose concentration difference between the inner and outer membrane slowed down the production of H^+ and the release of CO_2 , thereby exhibiting longer oscillation period in the subsequent cycles. The longest oscillation can exist for 960 min due to the completely consumed of sucrose in species A (Fig. 2d, Supplementary Fig .6). [...]

3) Secondly, the pH of a CO_2 saturated system should be more close to 4 at atmospheric pressure. It is therefore not logical that the pH increases that effectively.

Thank the reviewer for the comment. We measured the pH of pure water in the air for 60 h, indicating that the pH of air CO_2 saturated water solution is 5.8. However, with continuous purging CO_2 into the solution, the solution pH becomes 3.99, which is lower than that of air CO_2 saturated water solution. In terms of our solution (400 mM gadobutrol), the air CO_2 saturated pH value is 7.25 after continuous monitoring for 60 h. With purging CO_2 into our solution, the pH value is 5.21. Because the pH value of air CO_2 saturated gadobutrol solution is 7.25, the CO_2 produced by yeasts dissipate into the air, leading to an increase of solution pH.

4) The authors did not perform a control reaction in which they use only yeast cells and vary the sucrose concentration in the window that is achievable by the release from artificial cell A.

Thank the reviewer for the comment. We carried out the suggested experiments, and added below contents in pages 11 and 23.

[...] The faster CO_2 dissipation rate balanced CO_2 production rate, which explained the lower steady median pH value, the bigger amplitude and shorter average period at higher sucrose concentration in species A. In order to confirm the necessity of pH responsive release of sucrose from species A to the oscillation phenomenon, the solution pH variation of direct mixing sucrose with yeasts were measured (Supplementary Fig .10). The sucrose concentrations leaking out from species A (containing 50 mM, 100 mM, 150 mM, 200 mM, 250 mM, 300 mM sucrose) for 30 minutes (Supplementary Fig .10a) were chosen to mix with yeasts. No solution pH oscillations were observed (Supplementary Fig .10b). The solution pH dropped in the first phase, and gradually increased in the second phase.

[...] The prepared artificial cells containing 10 mmol/L pH-responsive molecule, sucrose (50 mM, 100 mM, 150 mM, 200 mM, 250 mM, 300 mM) and 0.05 g/mL PEG-20000 were suspended in 500 μ L isotonic solution of different pH with the addition of melittin (1 μ g/mL). Then the supernatant was collected by centrifugation at different time points. [...]

Supplementary Fig .10 (a) The sucrose concentration leaking out from species A (encapsulating different initial sucrose concentrations) as a function of time. **(b)** The solution pH variation over time caused by the mixing of yeasts and sucrose with the concentration after 30 minute leakage from species A (purple box in a). The sucrose concentration leaking out for 30 minute from species A containing 50 mM, 100 mM, 150 mM, 200 mM, 250 mM are 0.74 ± 0.05 mM, 1.45 ± 0.08 mM, 2.17 ± 0.12 mM, 3 ± 0.06 mM, and 3.6 ± 0.12 mM, respectively.

5) I find it difficult to see how the closely packed cell populations could prevent diffusion of small molecules or protons. The authors should study this in more detail with e.g. model dye components.

Thank the reviewer for the comment. We carried out dye diffusion experiments in different spatial coded communities using fluorescein as a model dye. We modified the main text as below in pages 16 and 24.

[...] The intensity of community A'CB (Fig. 4i) continuous increased, because the release of G6P from species A' at relative high pH (Fig. 4g) diffused into the adjacent species C to generate NADH. In order to observe the inhibitory effect of tightly packed species on the diffusion of small molecules more intuitively, we prepared a species A'' encapsulating model dye molecule of fluorescein (Mw=332.3 g/mol), which has green fluorescence and molecular weight similar to sucrose (Mw=342.3 g/mol) (Supplementary Fig .21a). Two three-species communities were constructed by arranging species A'', species B (Supplementary Fig .21b), and species C (labeled with TR DHPE, Supplementary Fig .21c) in the order of A''CB (Supplementary Fig .21d, e, and f) and CBA'' (Supplementary Fig .21h, i, and j). No green fluorescence was observed in the species B region in A''CB community within 270 minutes (Supplementary Fig .21g) due to the close pack of species C. Similarly, no green fluorescence was observed in the species C region due to the close pack of species B (Supplementary Fig .21k). All abovementioned results demonstrated that the spatial distribution of species in the community did exhibit dramatic impact on the behavior of community system.

[...] By varying the addition order of pH-responsive artificial cells containing sucrose and G6P (species A'), yeasts (species B) and artificial cells containing NAD⁺ and G6PDH (species C), A'CB, CA'B and CBA' were obtained. For dye penetration experiments, species A'' (containing G6P, fluorescein and sucrose), species B and species C were coded to obtain A''CB and CBA''. To prevent GUVs from breaking during the experiment, 300 μ L of DPPC ethanol-water solution (0.10 mg/mL, ethanol: water = 40 %:60 %, v/v) was added to coverslips, followed by heating at 60 $^{\circ}$ C for 10 min and washing 3 times with 400 mM gadobutrol solution. [...]

Supplementary Fig .21. (a) Species A'' encapsulating fluorescein and pH-responsive molecules. (b) Yeast (species B). (c) Species C labelled with TR DHPE. Schematic diagram of A''CB community with top view (d) and side view (e). (f) Merged image of laser scanning confocal images of A''CB community taken with red and green channels, and white field. (g) Laser scanning confocal microscopy images of A''CB community with green channel over time. Schematic diagram of CBA'' community with top view (h) and side view (i). (j) Merged image of laser scanning confocal images of CBA'' community taken with red and green channels, and white field. (k) Laser scanning confocal microscopy images of CBA'' community with green channel over time. The scale bars were 20 μm in a, b and c, and 100 μm in f, g, j, and k.

Reviewer #3 (Remarks to the Author):

Han et al. tried to construct unique artificial cells and create cell communities consisting of artificial cells and natural cells. The core technique to accomplish this research is the use of pH-sensitive molecule that phase-shift between fluid and gel phases, and encapsulate it inside giant vesicles. By changing the surrounding pH, the artificial cells release sugar into the environment and the neighbor yeasts consume the sugar and release CO₂ into the environment. As a consequence, the oscillation of pH change was generated in the society. The authors developed the society more complicated by increasing the species as 2 types of artificial cells and yeasts. Finally, the authors demonstrate the controlled communication between the species by giving spatial constraints.

The concept of an artificial community with artificial cells and natural cells is previously proposed by Mansy et al. (2014) and Stano et al. (2018). But this work is further advanced with fine-tuning. This would be a noteworthy paper in the field of artificial cell research which is rapidly developing nowadays.

The data are clear and analyzed well. Figures are well arranged to easily understand the results. But, unfortunately, the description in the Discussion is poor. The authors just repeat the explanation of what they did but mentioned nothing further. The authors should describe, at least, how the artificial system achieved in this work will benefit the study of biological phenomena, or how does it useful for controlling the biological system.

Therefore, before accepting the paper, the reviewer believes that the discussion part should be enriched so that the readers can understand what value this work has for life science as a whole.

Thank the reviewer for the comment. We added the descriptions to emphasize the significance of our system as below. The mentioned papers were cited in the manuscript.

Page 12

[...] The controlled release of G6P from species A' was expected to trigger the stepped generation of NADH inside species C (Fig. 3a). **Glucose-6-phosphate dehydrogenase (G6PDH) is an important enzyme involved in glycolysis pathway. Upon receiving G6P, the G6PDH inside species C converted G6P and NAD⁺ to 6-phosphate gluconate lactone and NADH, respectively. NADH is important biomolecule involved in many metabolic pathways. Species C is an artificial cell capable of metabolism mimicry.**

Page 13

[...] No NADH was generated inside species C with the absence of G6PDH in species C (Fig. 3e, blue curve), NAD⁺ in species C (Fig. 3e, black curve) or G6P in species A (Fig. 3e, red curve), which confirmed the feedback between species A' and species B regulated the internal 'metabolism' in species C. **The species A', B, C and environment were an integral system, among which these elements activities were influenced by others. We established a three-species-community with complex internal dynamic interactions, which helps to understand the behavior of complex community.**

Page 18

The construction of synthetic communities plays an important role in studying the interaction between species and the environment or understanding the complex behavior among populations. Three artificial cells are built by encapsulating sucrose/pH-responsive molecule (species A), sucrose/pH-responsive molecule/G6P (species A'), and NAD⁺ (species C), respectively. Yeasts are species B. A two-species community is constructed using species A and species B, which causes pH oscillation of its environment by the balance of sucrose consumption by yeasts and CO₂ dissipation from solution. The sucrose released from species A is consumed by yeast (species B) to generates CO₂ to decrease solution pH, whilst the oversaturated CO₂ dissipates into the air to increase solution

pH. The pH oscillation behavior is influenced by the initial sucrose concentration in species A and the ratio between species A and B. The higher initial sucrose concentration causes lower steady median pH values, bigger amplitude and shorter period. The higher ratio of species A to B results in lower steady median pH values, bigger amplitude and shorter period. Significantly, the pH oscillation between species A' and B in a three-species-community system regulates the periodical release of G6P from species A', which tunes the 'metabolic' reaction from NAD⁺ to NADH inside species C. The location of species in the spatial coded three-species-community system is confirmed to have dramatic impact on the interaction among species and between community and its environment. **The species in natural communities often exist in spatial orders. The influence study of spatial distribution on signal transmission among species in spatial coded synthetic communities helps to understand the structure and function of communities.**

The signal communication among species in nature communities has dynamic characteristics. By responding to environmental changes or signal molecules secreted by other species, species in the community can regulate their metabolism to better adapt to the environment and improve their survival ability. Two types of artificial cells were developed with the functions of environmental stimulus response (species A) and metabolism mimicry (species C). The environment as a key element was dynamically involved in the system to influence the function of species in constructed communities. The complex interactions among species and environment were demonstrated in the two-species/three-species communities. Therefore, we established communities possessing complex dynamic interaction network, which provide a way for investigating complicated inter network reaction among micro-ecosystems, and laid the foundation for building more complex systems in the future to perform higher-order functions.

References

25. Lentini, R. *et al.* Integrating artificial with natural cells to translate chemical messages that direct E-coli behaviour. *Nat. Commun.* **5**, 4012 (2014).
28. Rampioni, G. *et al.* Synthetic cells produce a quorum sensing chemical signal perceived by *Pseudomonas aeruginosa*. *Chem. Commun.* **54**, 2090-2093 (2018).

The following are questions and minor revisions.

1) Page 6, Fig. 1, Is it possible to know the rate of release of the sucrose from the vesicles at pH 6.6?

Thank the reviewer for the comment. The release rate of the sucrose from vesicles depends on the concentration difference between inside and outside vesicles. Therefore, it is a varying value. The average of sucrose release rate from vesicle (containing 300 mM sucrose) at pH 6.6 was 0.094 ± 0.003 mM/min.

2) Page 9, Fig. 2, Did the yeast proliferate during the cycle turning?

Thank the reviewer for the comment. We monitored the density of yeasts every 150 min for 600 minutes (Fig. R5). No significant proliferation was observed.

Fig. R5. The variation of yeast population density during oscillation process

3) Page 9, Fig. 2, Revise the head title of Fig. 2.

Thank the reviewer for the comment. We modified the head title of Fig. 2 as below.

Fig. 2. Construction of two-species-community of pH-responsive artificial cells (containing sucrose) (species A) and yeast (species B) and its pH oscillatory environment. **a**, Schematic illustration of solution pH oscillation caused by the feedback between species A and B. **b**, A typical solution pH oscillation of two-species-community system as a function of time. Species A and B was $2.76 \times 10^6/\text{mL}$ and $4.36 \times 10^6/\text{mL}$ respectively. **c**, pH value of CO_2 oversaturated (phase I, red curve) and saturated (phase II, blue curve) gadobutrol solution as a function of time. CO_2 was injected into gadobutrol solution in phase I. The gadobutrol solution was saturated by CO_2 from air in phase II. **d**, The initial sucrose concentration and sucrose concentration at the end of oscillation after 960 min. The sucrose concentrations were obtained by adding 10 % Triton-100 into the solution to release sucrose from species A. Data are presented as mean values \pm SD, $n = 3$. **e**, Flow cytometry scatter plots of live-dead stained species B at longest oscillation time (960 min). Live yeasts (species B) were stained with FDA (green, in Q3), and dead yeasts were stained with PI (red, in Q1). Total number of particles counted was 10,000.

4) Page 11, line 175, “7.2)” -> “7.2”

Thank the reviewer for the comment. We have corrected “7.2)” to “7.2” as below in page 11.

[...] The CO_2 dissipation rate was faster at lower pH, since the slopes from the first lowest point to the first highest point were bigger at higher sucrose concentration (Fig. 3b, purple dashed boxes) and the tangents of the points in the red curve ranging from 6.5 to 7.2 (Fig. 2d, phase I) were bigger. [...]

5) Page 13, line 217, “Fig. 3C”. This might be “Fig. 4c”.

Page 13

Thank the reviewer for the comment. We have corrected “Fig. 3C” to “Fig. 3c” in the revised manuscript.

6) Page 13, line 227, “Fig. 3d”. This might be “Fig. 4c”.

Page 13

Thank the reviewer for the comment. We have corrected “Fig. 3d” to “Fig. 3c” in the revised manuscript.

7) G. Rampioni et al. (*Chemical Communications* 2018) also reported pioneering work in this research line. This should be cited in the text.

Thank the reviewer for the comment. We have cited this paper as below.

28. Rampioni, G. *et al.* Synthetic cells produce a quorum sensing chemical signal perceived by *Pseudomonas aeruginosa*. *Chem. Commun.* **54**, 2090-2093 (2018).

Reviewers' Comments:

Reviewer #1:

Remarks to the Author:

There are interesting aspects to this work, but the writing and presentation are poor. While some of this is a bit unfair as it may reflect different native languages, I am worried that the poor presentation may additionally reflect a lack of thought as to the implications of the work.

For example, a response in the rebuttal that it's OK to use the term artificial cell to describe their system since others have called similar structures artificial cells is not logically sound. It's no big deal to me if the authors call their structures artificial cells (even if I don't think they are), but there are plenty of examples of people doing or saying wrong things. I would have preferred a logical argument justifying their claim.

I had asked the authors to name their pH responsive molecule. They now have in the methods section. This isn't really enough for the reader to easily follow the story. No explanation is given (or reference for that matter) for the choice of this molecule. How did they know to synthesize this molecule? Also, is there evidence for the H-bonding pattern shown in figure 1a?

There seems to be no reason for the 3rd "artificial cell" that contains NAD. The authors argue in their rebuttal that this artificial cell is important because it has an enzyme important in the glycolysis pathway (glucose-6-phosphate dehydrogenase) and then go on to claim that "The species A, B, C and environment were an integral system," No data demonstrated that "C" was integral to the system. I personally think that the NAD containing artificial cell should be removed from the story. However, if this artificial cell were only used to demonstrate the effect of spatial arrangement and not to claim that they've built a more complex community of artificial cells, then it would make much more sense. This 3rd artificial cell is not providing any useful function.

How sucrose was quantified was not explained. Just mentioning that an enzyme coupled assay was used would make following the text easier.

Citing references 28 and 29 for artificial cells that engage in quorum signaling is strange since the cited papers were published in 2018 and 2019. Two-way chemical communication through quorum signaling with artificial cells and natural cells was shown earlier in 2017 (DOI: 10.1021/acscentsci.6b00330).

I'm sure my comments sound more negative than I actually feel about the science. There is something interesting here, and I don't doubt the science. The presentation is just poor. The introduction states "The dynamic interactions between different species and their environment can improve their opportunity of survival by showing more advanced overall behavior through the cooperative operation of individual cells in compare with single cell." It would have been nice if the authors explained to the reader how their work brought us closer to that goal.

Reviewer #2:

Remarks to the Author:

The authors have performed additional experiments and provided further explanations of their system. Still a number of significant questions remain.

Based on the changing sucrose concentrations in the artificial cells and changing release profiles as shown in SI fig 10 the oscillations should also diminish over time (as one in situ changes from a high to low sucrose concentration in the artificial cells). This is not observed in the oscillation patterns of e.g. SI fig 9. Furthermore, at a certain moment, if the system resets itself by loss of CO₂ to the environment, this process should take over and not allow the pH to be decreased at lower sucrose

conditions. The authors should explain this more effectively. They should show the oscillation pattern until full sucrose conversion.

The limited diffusion in the close packed systems should not be studied by encapsulated dyes; the authors should add dyes in solution, or demonstrate that the dyes can be released from the artificial cells.

Reviewer #3:

Remarks to the Author:

The authors have sufficiently addressed my comments.

Response to the reviewers' comments

For the sake of clarity, the comments of the reviewer have been collated in black, and our response to each comment appears in blue. All the changes to the manuscript are highlighted in red.

Reviewer #1 (Remarks to the Author):

1) There are interesting aspects to this work, but the writing and presentation are poor. While some of this is a bit unfair as it may reflect reflect different native languages, I am worried that the poor presentation may additionally reflect a lack of thought as to the implications of the work.

Thank the reviewer for the comment and valuable suggestion. We improved the writing and presentation by professional language editing service “Nature publishing group language editing service”. The editing certificated is provided at blow.

2) For example, a response in the rebuttal that it's OK to use the term artificial cell to describe their system since others have called similar structures artificial cells is not logically sound. It's no big deal to me if the authors call their structures artificial cells (even if I don't think they are), but there are plenty of examples of people doing or saying wrong things. I would have preferred a logical argument justifying their claim.

Thank the reviewer for the comment and valuable suggestion. We fully agree with your viewpoint. “Artificial cells” are defined as the cell-like structures which mimic partial (or whole) cell function and structures in this field. Species A is a cell-like structure containing viscous gel similar to cytoplasm and melittin-embedded lipide bilayer membrane similar to cell membrane to some extents. Species C is a cell-like structure containing G6PDH to catalyzing G6P and NAD⁺ molecules to produce NADH inside melittin-embedded lipid bilayer membrane, which can mimic metabolism process of cell to some extents. We added below sentences in page 3.

[...] Understanding these complex dynamic behaviours will help us decode the operating principles of biological systems that support and maintain life; in addition, this knowledge will provide a foundation for researchers to advance future microscale technologies that exhibit key features of living systems.⁶⁻¹¹ Artificial cells are cell-like structures that mimic partial/whole cell structure and functions. These cells have been prepared using natural or synthetic materials¹²⁻¹⁴ and are used to clarify the working mechanism of cells.¹⁵⁻²² [...]

3) I had asked the authors to name their pH responsive molecule. They now have in the methods section. This isn't really enough for the reader to easily follow the story. No explanation is given (or reference for that matter) for the choice of this molecule. How did they know to synthesize this molecule? Also, is there evidence for the H-bonding pattern shown in figure 1a?

Thank the reviewer for the comment. In this paper, a pH responsive gel is needed to couple the pH variation of dissolving and dissipating of CO₂. There are many pH responsive gels including polymer gels (Curr. Med. Chem., 2020, 27, 2631; Int. J. Pharm., 2021, 592, 120047; Eur. Polym. J., 2022, 177, 111473; Molecules, 2023, 28, 4246; Gels, 2021, 7, 68) and molecular gels (Soft Matter, 2017, 13, 1914; Chem. Soc. Rev., 2021, 50, 5165; Langmuir, 2005, 21, 109; Org. Biomol. Chem., 2015, 13, 561-569; Chem, 2017, 3, 390; Soft Matter, 2018, 14, 6166) in the literatures. Among them, the molecular gels respond to pH changes more quickly (Soft Matter, 2009, 5, 1856; Langmuir, 2009, 25, 8639; Angew. Chem. Int. Ed., 2015, 54, 13258; Angew. Chem. Int. Ed., 2021, 60, 9973; Front. Chem., 2021, 9, 770102) The pH responsive molecular gel herein possesses fast response property and suitable pH response range. We added below sentences in page 4.

Although there are many pH-responsive hydrogelation molecules³⁵⁻³⁹, the pH-responsive molecule here possesses fast response properties and a suitable pH response range.

We carried out ATR-FTIR tests on the hydrogels at pH=8 and pH=5 to prove the hydrogen bonds in the hydrogels. We added below sentences in page 4.

ATR-FTIR data confirmed the presence of hydrogen bonds in the hydrogels (Supplementary Fig. 2). At pH 8, the peak at 1550 cm⁻¹ is caused by C-N stretching and N-H bending⁴⁰, and the peak at 1400 cm⁻¹ corresponds to the stretching vibration of the carboxyl Group⁴¹. At pH 5, these two peaks shifted to 1540 cm⁻¹ and 1390 cm⁻¹, respectively, which results from the formation of hydrogen bonds among molecules in gel phase.⁴²

Supplementary Fig .2 ATR-FTIR spectra at different pH conditions

4) There seems to be no reason for the 3rd "artificial cell" that contains NAD. The authors argue in their rebuttal that this artificial cells is important because it has an enzyme important in the glycolysis pathway (glucose-6-phosphate dehydrogenase) and then go on to claim that "The species A', B, C and environment were an integral system," No data demonstrated that "C" was integral to the system. I personally think that the NAD containing artificial cell should be removed from the story. However, if this artificial cell were only used to demonstrate the effect of

spatial arrangement and not to claim that they've built a more complex community of artificial cells, then it would make much more sense. This 3rd artificial cells is not providing any useful function.

Thank the reviewer for the comment and valuable suggestion. We fully agree with your viewpoint. We have removed the sentence of "The species A', B, C and environment were an integral system, ..." and emphasized the significance of species C in demonstrating the effect of spatial distribution. We have modified the text as below in page 14.

[...] No NADH was generated inside species C with the absence of G6PDH in species C (Fig. 3e, blue curve), NAD⁺ in species C (Fig. 3e, black curve) or G6P in species A (Fig. 3e, red curve), which confirmed that the feedback between species A' and species B regulated the internal metabolism in species C. **In the following context, this phenomenon was used to investigate the effect of species spatial distribution on their communications.**

5) How sucrose was quantified was not explained. Just mentioning that an enzyme coupled assay was used would make following the text easier.

Thank the reviewer for the comment. A sucrose detection kit was used to quantitatively detect sucrose. The kit was obtained from Nanjing Jiancheng Bioengineering Institute (China). Sucrose was hydrolysed in an acidic hydrolysate to produce glucose and fructose. Fructose further converts into 5-hydroxymethylfurfural in an acidic hydrolysate, which possesses a maximum absorption wavelength of 290 nm. Therefore, by monitoring the absorbance of the solution at 290 nm together with the sucrose calibration curve, the quantitative detection of sucrose was achieved. We added below sentences in page 23.

[...] Sucrose release was tested using a sucrose kit. Specifically, 30 µL of supernatant and 2 mL of hydrolysed solution were thoroughly mixed and subsequently heated in a water bath at 100 °C for 8 min. **Sucrose was hydrolysed in an acidic hydrolysate to produce glucose and fructose. Fructose further converts into 5-hydroxymethylfurfural in an acidic hydrolysate, which possesses a maximum absorption wavelength of 290 nm. Therefore, by monitoring the absorbance of the solution at 290 nm together with the sucrose calibration curve, the quantitative detection of sucrose was achieved.**

6) Citing references 28 and 29 for artificial cells that engage in quorum signaling is strange since the cited papers were published in 2018 and 2019. Two-way chemical communication through quorum signaling with artificial cells and natural cells was shown earlier in 2017 (DOI: 10.1021/acscentsci.6b00330).

Thank the reviewer for the comment. We added the reference (2017) in the quorum sensing position as below.

[...] Through the secretion and recognition of diffusive signalling molecules in their local environment, artificial cells/live cells in synthetic communities can actively communicate with each other and their surrounding environment to realize critical dynamic biological behaviours, such as predation²³, protein expression^{24, 25}, motility^{26, 27}, quorum sensing²⁸⁻³⁰, and differentiation^{31, 32}. [...]

28. **Lentini, R. *et al.* Two-way chemical communication between artificial and natural cells. *ACS Central Sci.* **3**, 117-123 (2017).**
29. Rampioni, G. *et al.* Synthetic cells produce a quorum sensing chemical signal perceived by *Pseudomonas aeruginosa*. *Chem. Commun.* **54**, 2090-2093 (2018).

30. Rampioni, G., Leoni, L. & Stano, P. Molecular communications in the context of "synthetic cells" research. *IEEE T. NanoBiosci.* **18**, 43-50 (2019).

7) I'm sure my comments sound more negative than I actually feel about the science. There is something interesting here, and I don't doubt the science. The presentation is just poor. The introduction states "The dynamic interactions between different species and their environment can improve their opportunity of survival by showing more advanced overall behavior through the cooperative operation of individual cells in compare with single cell." It would have been nice if the authors explained to the reader how their work brought us closer to that goal.

Thank the reviewer for the comment. Individual species A or B cannot cause pH oscillation of the solution. The cooperation between species A and B led to pH oscillation of the solution, which kept the pH variation of the entire solution within a small range, maintaining dynamic pH stability of the system. If a species is very sensitive to the solution pH, the system pH stability will improve its survival opportunity. In that sense, our system holds the potential to improve species survival opportunity to some extents. We added below sentences in page 12.

[...] The construction of a two-species-community oscillation system helps researchers construct complex dynamic response systems using simple components. **The dynamic interaction among species A, species B and the environment maintained the pH stability of the system, which may be beneficial to the survival of pH-sensitive species.** [...]

Reviewer #2 (Remarks to the Author):

The authors have performed additional experiments and provided further explanations of their system. Still a number of significant questions remain.

1) Based on the changing sucrose concentrations in the artificial cells and changing release profiles as shown in SI fig 10 the oscillations should also diminish over time (as one in situ changes from a high to low sucrose concentration in the artificial cells). This is not observed in the oscillation patterns of e.g. SI fig 9. Furthermore, at a certain moment, if the system resets itself by loss of CO₂ to the environment, this process should take over and not allow the pH to be decreased at lower sucrose conditions. The authors should explain this more effectively. They should show the oscillation pattern until full sucrose conversion.

Thank the reviewer for the comment. We followed the suggestion and carried out experiment to show the oscillation pattern until full sucrose conversion. We added below sentences in page 11.

[...] The faster CO₂ dissipation rate balanced the CO₂ production rate, which explained the lower steady median pH value, the larger amplitude and shorter average period at higher sucrose concentrations in species A. The gradual consumption of sucrose from species A caused pH oscillation to disappear since the CO₂ production rate was not enough to overcome the CO₂ dissipation rate.

Supplementary Fig .10 (c) The solution pH oscillation of two-species-community system as a function of time with 200 mM sucrose concentration in species A. ...

2) The limited diffusion in the close packed systems should not be studied by encapsulated dyes; the authors should add dyes in solution, or demonstrate that the dyes can be released from the artificial cells.

Thank the reviewer for the comment. We followed the suggestion and carried out experiments of the dye-releasing from individual artificial cell and spatial patterned artificial cells. The leakage of fluorescein over time were observed both from individual artificial cell and spatial patterned artificial cells. We added below sentences in page 16.

[...] To observe the inhibitory effect of tightly packed species on the diffusion of small molecules more intuitively, we prepared species A'' encapsulating a model dye molecule of fluorescein (Mw=332.3 g/mol), which exhibits green fluorescence and a molecular weight similar to sucrose (Mw=342.3 g/mol). **Fluorescein leakage was observed from individual species A'' (Supplementary Fig. 22) and spatially patterned A'' (Supplementary Fig. 23).** Two three-species communities were constructed by arranging species A'' (Supplementary Fig. 24a), species B (Supplementary Fig. 24b), and species C (labelled with TR DHPE, Supplementary Fig. 24c) in the order of A''CB (Supplementary Fig. 24d, e, and f) and CBA'' (Supplementary Fig. 24h, i, and j). [...]

Supplementary Fig. 22 (a, b) Schematic illustration of fluorescein leakage over time from artificial cell A'' containing pH-responsive molecules, sucrose, G6P, and fluorescein. **(c, d)** The corresponding fluorescence microscope images of A'' at 0 min and 270 min with solution pH 7.2. **(e)** The line profiles of the white dashed line in (c, d). The scale bars were 5 μm.

Supplementary Fig. 23 (a, b) Schematic illustration of fluorescein leakage over time from spatially patterned A'' by magnetic field. **(c, d)** The corresponding fluorescence microscope images of spatially patterned A'' at 0 min and 270 min with solution pH 7.2. **(e)** The line profiles of the white dashed line in (c, d). The scale bars were 200 μm.

Reviewers' Comments:

Reviewer #1:

Remarks to the Author:

I am satisfied with the revisions and have no further comments to make.

Reviewer #2:

Remarks to the Author:

the authors have performed the requested additional experiments. The manuscript is now suitable for publication

Response to the reviewers' comments

For the sake of clarity, the comments of the reviewer have been collated in black, and our response to each comment appears in blue.

Reviewer #1 (Remarks to the Author):

1) I am satisfied with the revisions and have no further comments to make.

We really appreciated the comments from the reviewer to help us improve the quality of our paper.

Reviewer #2 (Remarks to the Author):

1) the authors have performed the requested additional experiments. The manuscript is now suitable for publication

We really appreciated the comments from the reviewer to help us improve the quality of our paper.